# Molecular Phylogeny, Morphology, Growth and Toxicity of Three Benthic Dinoflagellates *Ostreopsis* sp. 9, *Prorocentrum lima* and *Coolia monotis* Developing in Strait of Gibraltar, Southwestern Mediterranean

**DOI:** 10.3390/toxins16010049

**Published:** 2024-01-16

**Authors:** Mustapha Ibghi, Benlahcen Rijal Leblad, Mohammed L’Bachir El Kbiach, Hicham Aboualaalaa, Mouna Daoudi, Estelle Masseret, Emilie Le Floc’h, Fabienne Hervé, Gwenael Bilien, Nicolas Chomerat, Zouher Amzil, Mohamed Laabir

**Affiliations:** 1Marine Environment Monitoring Laboratory, INRH (Moroccan Institute of Fisheries Research), Tangier 90000, Morocco; mustaphaibghi2015@gmail.com (M.I.); hicham.aboualaalaa@etu.uae.ac.ma (H.A.); daoudi@inrh.ma (M.D.); 2Equipe de Biotechnologie Végétale, Faculty of Sciences, Abdelmalek Essaadi University Tetouan, Tetouan 93000, Morocco; melkbiach@uae.ac.ma; 3MARBEC, University of Montpellier, CNRS, IRD, Ifremer, 34095 Montpellier, France; estelle.masseret@umontpellier.fr (E.M.); emilie.lefloch@cnrs.fr (E.L.F.); 4Laboratoire Phycotoxines, IFREMER (French Research Institute for Exploitation of the Sea)/PHYTOX/METALG, 44311 Nantes, France; fabienne.herve@ifremer.fr (F.H.); zouher.amzil@ifremer.fr (Z.A.); 5IFREMER, Unité Littoral, Station de Biologie Marine, Place de la Croix, 29185 Concarneau, France; gwenael.bilien@ifremer.fr (G.B.); nicolas.chomerat@ifremer.fr (N.C.)

**Keywords:** benthic harmful algal blooms, southwestern Mediterranean, *Prorocentrum lima*, *Coolia monotis*, *Ostreopsis* sp. 9, growth, toxins

## Abstract

Few works have been carried out on benthic harmful algal blooms (BHAB) species in the southern Mediterranean and no data are available for the highly dynamic Strait of Gibraltar (western Mediterranean waters). For the first time, *Ostreopsis* sp. 9, *Prorocentrum lima* and *Coolia monotis* were isolated in this key region in terms of exchanges between the Atlantic Ocean and the Mediterranean and subject to intense maritime traffic. Ribotyping confirmed the morphological identification of these three dinoflagellates species. Monoclonal cultures were established and the maximum growth rate and cell yield were measured at a temperature of 24 °C and an irradiance of 90 µmol photons m^−2^ s^−1^, for each species: 0.26 ± 0.02 d^−1^ (8.75 × 10^3^ cell mL^−1^ after 28 days) for *Ostreopsis* sp. 9, 0.21 ± 0.01 d^−1^ (49 × 10^3^ cell mL^−1^ after 145 days) for *P. lima* and 0.21 ± 0.01 d^−1^ (10.02 × 10^3^ cell mL^−1^ after 28 days) for *C. monotis*. Only *P. lima* was toxic with concentrations of okadaic acid and dinophysistoxin-1 measured in optimal growth conditions ranging from 6.4 pg cell^−1^ to 26.97 pg cell^−1^ and from 5.19 to 25.27 pg cell^−1^, respectively. The toxin content of this species varied in function of the growth phase. Temperature influenced the growth and toxin content of *P. lima*. Results suggest that future warming of Mediterranean coastal waters may lead to higher growth rates and to increases in cellular toxin levels in *P. lima*. Nitrate and ammonia affected the toxin content of *P. lima* but no clear trend was noted. In further studies, we have to isolate other BHAB species and strains from Strait of Gibraltar waters to obtain more insight into their diversity and toxicity.

## 1. Introduction

Given their impacts on human health and on the fish and mollusk farming industry, many studies have been carried out on the distribution and the proliferation of benthic harmful algal bloom (BHAB) dinoflagellates [1,2,3,4,5,6,7,8,9]. The development of BHAB species has been linked to anthropogenic impacts and ocean warming [7,10]. Numerous BHAB species, renowned for their thermophily, are increasingly proliferating within temperate marine ecosystems [11,12,13,14,15,16,17]. These dinoflagellates have been documented in the Mediterranean Sea, with water temperature being the main environmental factor controlling their development [18,19]. In the Mediterranean, the most abundant benthic dinoflagellate species detected were *Ostreopsis* cf. *ovata*, *Ostreopsis* cf. *siamensis*, *Coolia monotis* and *Prorocentrum lima*. Numerous studies have examined the environmental parameters influencing the blooms of BHAB species in the Mediterranean Sea [8,15,17,20,21,22,23,24,25,26,27,28,29,30,31,32,33,34]. Reports indicated that the development of *Ostreopsis* is on the rise in the Mediterranean [30,35,36,37,38]. *Ostreopsis* species produce complex toxic non-peptide compounds from the palytoxin (PLTX) group [39]. Cases of human intoxication by PLTXs were documented in the Mediterranean (Spain in 2004, Italy in 2005 and 2006, France from 2006 to 2009, Algeria in 2009) [26,40,41,42,43] and in Atlantic Sea [43,44]. *O*. *siamensis* has been documented to thrive in tropical and subtropical ecosystems around the world, such as the Gulf of Thailand [45], the Pacific Ocean, including Ryukyu Islands and French Polynesia [46,47], and the Caribbean region (specifically in Belize and Guadeloupe Island) [48,49]. A different species named *O*. cf. *siamensis* has been described in various temperate and subtropical regions. This species has been reported in locations like Portugal [50,51], Greece [52], Italy [35], the Moroccan Atlantic Ocean [53,54], New Zealand [55,56], Spain [35,57], Lebanon [58], Tunisia [59] and France [43]. While *O*. *siamensis* has been shown to produce toxins such as ostreocins [46,60], no specific toxin has been identified in *O*. cf. *siamensis.* However, Verma et al. [61] reported that the PLTX-like compounds were produced by a strain of *O.* cf. *siamensis*, as determined by chemical analysis, and the LD_50_ of the cell extract determined by intraperitoneal injection in mice was 25.1 mg kg^−1^. In both Mediterranean and Atlantic waters, *O*. cf. *siamensis* does not produce any recognized toxins, unlike *O*. cf. *ovata* [36,41,43,62,63,64], leaving its toxigenic status still uncertain. 

The epiphytic/epibenthic *Prorocentrum lima* (Ehrenberg) F. Stein 1878 [65] is a cosmopolitan species which was described in several marine regions in the Mediterranean [19,66,67,68]. *P. lima* can be found associated with macrophytes, rocks, floating detritus and debris as substrates. This species is responsible for the diarrheic shellfish poisoning (DSP) syndrome [69,70]. *P. lima* produces toxins such as okadaic acid (OA) and dinophysistoxins (DTXs), which cause symptoms including nausea, abdominal pain, diarrhea and emesis in consumers [66,67,71,72,73]. Lipophilic shellfish toxins (LSTs), OA and DTXs are potent inhibitors of protein phosphatases 2A, 1B and 2B, which may promote cancer in the human digestive system [74,75]. The *P. lima* species complex is abundant and distributed worldwide [76]. *P. lima* was observed in marine ecosystems of numerous countries including Canada [77], USA [68,78], Brazil [79,80], Venezuela [81], Argentina [82], Cuba [83], United Kingdom [69,84], Portugal [85], Spain [57] and Sweden [86]. In the Mediterranean Sea, this species has been reported in Greece [52], France [29,87], Tunisia [21,30,32,88], Italy [65] and Morocco [34]. *P. lima* has also been identified in other regions such as the NW Adriatic coastline [89] and Pacific waters including Japan, Okinawa, Ishigaki Island [90,91], Tahiti Island [66] New Colombia [92], Taiwan [93] and China, specifically in Beihai, Guangxi in the Chinese Sea [94] and along the coasts of Hong Kong and Hainan Island in Sanya [95].

*Coolia* Meunier is a genus of benthic dinoflagellates. *C. monotis* was first described in northern European waters by Meunier [96]. This species was observed worldwide, including in the Mediterranean Sea [97]. While earlier research indicated that *C. monotis* might produce toxic compounds known as cooliatoxins [98], recent studies have suggested that the species is either non-toxic or encompasses non-toxic strains [8]. As of now, there have been no reported poisoning incidents linked to *C. monotis*.

The Strait of Gibraltar, which connects the Mediterranean Sea to the Atlantic Ocean, is a pivotal marine corridor. This narrow channel frequently sees the passage of large ships, especially with the presence of major ports on both its flanks: Tangier Med in Morocco and Algeciras in Spain. Consequently, the strait can be exposed to ballast water discharges, with the potential introduction of toxic microalgae. The rising water temperatures due to climate change further exacerbate the vulnerability of the region, making it more prone to the potential installation of non-native species, including BHAB species [99,100,101,102].

There have been several intoxication episodes in the Western Moroccan Mediterranean due to different HAB species producing paralytic shellfish toxins (PSTs), amnesic shellfish toxins (ASTs) and LSTs. These toxins and related intoxications were associated with proliferations of *Gymnodinium catenatum*, *Pseudo-nitzschia* spp., *Dinophysis* spp. and *Prorocentrum lima*, respectively [103,104,105,106,107]. Ibghi et al. [34] documented the presence of *Ostreopsis* spp., *C. monotis* and *P. lima* during a one-year field study carried out in three sites covering the southern part of the Strait of Gibraltar. In addition, OA and DTX-1 were detected in the sampled natural phytoplankton [34]. In the Strait of Gibraltar (Cap Malabata, Oued Lihoud and Dalia locations), *P. lima* was observed between May 2019 and November 2020 with cell densities reaching up to 8.2 × 10^3^ cells g^−1^ of macrophytes fresh weight and 10^3^ cells L^−1^ in the water column.

The aim of this study was to comprehensively characterize, for the first time, three BHAB dinoflagellates (*Ostreopsis* sp. 9, *P. lima*, *C. monotis*) isolated in the southern part of the Strait of Gibraltar, focusing on their morphology, phylogeny and toxicity. Furthermore, we examined the impact of increasing temperature and varying concentrations of ammonia and nitrate on the growth, biometry and toxin levels within *P. lima* cultures.

## 2. Results

### 2.1. Culture Characteristics and Morphology

The cells of *Ostreopsis* sp. 9 OSCM17 strain were motile and exhibited size variability, with the emergence of small, thin and obscure cells across various culture phases corresponding to temporary cysts. Cells could be embedded in mucus and adhered to the wall of a flask particularly during the end of the exponential phase (Figure 1B). Cells of this dinoflagellate species were ovoid with a pointed, teardrop shape. Each cell housed several yellow–brown chloroplasts centrally located. The cells displayed rotational movement around their dorsal–ventral axis. The plate pattern was Po, 3’ 7’’ 6C, 6S (?) 5’’’ 2’’’’ 1p (Figure 1C,D). The apical pore (Po) measured 11.2 ± 1.3 µm in length (*n* = 32) and was situated on the dorsal left side of the epitheca (Figure 1D). The 1’ plate, which was large, hexagonal, relatively narrow and elongated, touched the 2’, 3’, 2’’, 6’’ and 7’’ plates (Figure 1C). The 3’ plate, pentagonal in shape, was positioned towards the left dorsal side of the epitheca, making contact with plates 1’, 4’, 5” and 6”. The apical plate 2’ was short and nearly matched the length of the Po plate. During the exponential phase, the cell length of the OSCM17 strain varied between 38.1 and 59.0 µm, averaging 48.8 ± 3.4 µm. The width ranged from 28.7 to 43.7 µm with a mean of 35.8 ± 3.6 µm. During the stationary phase, the cell length ranged from 46.0 to 60.8 µm, averaging 53.2 ± 3.0 µm, while the width ranged from 30.0 to 40.6 µm, with an average of 36.2 ± 2.4 µm (Table 1).

*P. lima* PLCM17 strain showed non-motility to weakly motile cells and formed dense, greenish mats at the bottom of the flask (Figure 2A). The growth of cultured cells lasted 145 days, exhibiting typical vegetative cells throughout the entire growth duration. The cells were oblong to oval; they widened in the central area and narrowed at the anterior end. On the right valve, the periflagellar area was V-shaped and featured a brown–golden chloroplast (Figure 2B). The cells exhibited pores on both valves, except in the cell’s central part (Figure 2D). During the exponential phase, the cell length fluctuated between 37.0 and 47.8 µm, averaging 42.9 ± 1.8 µm. The width averaged 31.3 ± 1.3 µm, with values ranging from 28.2 to 35.1 µm. In the stationary phase, the cell length varied between 40.3 and 48.3 µm, averaging 44.2 ± 1.8 µm, and the width ranged from 29.0 to 35.4 µm, with a mean of 32.5 ± 1.4 µm. Notably, while the length at the stationary phase contrasted, the width of *P. lima* cells closely resembled that during the exponential phase of growth (Table 1).

Cells of *C. monotis* CMON15 strain displayed rotational movements and were highly motile. Cells of this dinoflagellate were small, compressed, rounded and lens-shaped, containing a golden–brown chloroplast. An elongated, curved nucleus was situated in the dorsal zone (Figure 3A,B). The apical pore was elongated, located on the theca 2’. The plate formula was Po 3’ 7” 6C ? S 5”’ 2”’’ (Figure 3D). From the end of the exponential phase through to the decline, *C. monotis* cells excreted mucus into the culture medium, which was observable under a microscope (Figure 3B). During the exponential growth phase, the cell length of *C. monotis* varied between 30.2 and 38.1 µm, averaging 34.9 ± 2.0 µm, while the width ranged from 28.9 to 36.6 µm with an average of 32.8 ± 2.1 µm. In the stationary phase, the length ranged from 30.2 to 39.5 µm, with a mean of 34.1 ± 2.6 µm, and the width varied from 28.0 to 38 µm, averaging 32.3 ± 2.4 µm. No significant differences in cell sizes were noted when comparing the exponential and stationary growth phases (Table 1).

### 2.2. Molecular Analysis and Phylogeny

Sequences of 930, 856 and 887 base pairs of the partial large subunit (LSU) rDNA (D1–D3) were obtained from *Ostreopsis* sp. 9 (OSCM17), *P. lima* (PLCM17) and *C. monotis* (CMON15) strains, respectively. They were deposited in GenBank with accession numbers OR734855, OR734226 (OSCM17), OR734081 (PLCM17) and OR734856, OR734294 (CMON15). An ML phylogenetic tree of Ostreopsidoideae (Figure 4) showed that sequences of OSCM17 and CMON15 clustered in well-resolved clades of *Ostreopsis* sp. 9 (‘*Ostreopsis* cf. *siamensis’*) and *Coolia monotis*, respectively. The sequence of the OSCM17 strain was closely related to sequences of strains CSIC-D, CAWD203, HER24 and CAWD173 from the Catalan Sea and Australia, respectively. The sequence CMON15 was similar to sequences of strains Dn26EHU, CMBZT14, OCH228 and Com16 from Spain (Atlantic Ocean), Tunisia and the Canary Islands (Figure 4). As shown by the ML tree of the *Prorocentrum lima* complex and closely related species (Figure 5), the sequence of the PLCM17 strain clustered within clade 4 of the *P. lima* complex with high support. Other sequences included in this clade (Dn35EHU, Dn37EHU, Dn38EHU, Sorrento1, PLLS01, NCMA685) originated from Spain (Atlantic Ocean), Italy and Australia.

### 2.3. Toxins

Ovatoxins, PLTXs and cooliatoxins were analyzed in the cells harvested during the exponential and stationary growth phases of *Ostreopsis* sp. 9 strain OSCM17 and *C. monotis* strain CMON15, respectively. None of these toxins were detected. 

In contrast, *P. lima* PLM17 strain showed the presence of okadaic acid (OA) and DTX-1 (Figure 6). 

### 2.4. Growth Characteristics

After 19 days of culture, the maximum reached cell density (cell yield) of *Ostreopsis* sp. 9 was 8.7 × 10^3^ cells mL^−1^ (Figure 7A) with a maximum growth rate of 0.26 ± 0.02 d^−1^ (Figure 8). The growth curve displayed an exponential phase with an initial slow growth between the 1st and 9th days, followed by a more rapid growth phase between the 9th and 19th days. This culture did not show a prolonged stationary phase; the cell density dropped following the exponential growth. No lag phase was observed (Figure 7A). 

The growth curve of *P. lima* was marked by very slow growth lasting 100 days, followed by a prolonged exponential phase that continued for over 25 days (Figure 8B). The cell yield reached 49.5 × 10^3^ cells mL^−1^ after 145 days of culture (Figure 7B), and the maximum growth rate was 0.21 ± 0.016 d^−1^ (Figure 8). 

For *C. monotis*, the exponential phase began immediately after the inoculation and lasted for 28 days. A decline in the culture was observed from the 31^st^ day onwards (Figure 7C). The maximum cell density of this species reached 10^4^ cells mL^−1^ after 28 days (Figure 7C), with a growth rate of 0.21 ± 0.01 d^−1^ (Figure 8). 

### 2.5. Effect of Temperature on Toxin Content of Prorocentrum lima

The concentrations of LSTs produced by *P. lima* PLCM17 strain varied depending on the growth phase and the temperature tested (Figure 9, Table 2). During the exponential phase, the maximum values were observed at 15 °C (11.63 pg cell^−1^ for OA and 15.27 pg cell^−1^ for DTX-1). In contrast, during the stationary phase, the maximum LST concentrations were registered at 24 °C (26.97 pg cell^−1^ for OA, and 25.27 pg cell^−1^). Growth rates were higher at 20 °C and 24 °C, with 0.22 and 0.21 d^−1^, respectively. Growth decreased to 0.08 and 0.06 d^−1^ for cells cultured at 15 °C and 29 °C, respectively. Cell width had an opposite trend, with cells being larger at 15 °C and 29 °C in comparison to those cultivated at 20 °C and 24 °C (Table 2, Figure 10).

### 2.6. Effect of Nitrogen Source on Toxin Content of Prorocentrum lima 

For nitrogen experiments, cultures were grown at a temperature of 24 °C. Data showed that LST concentrations were slightly higher in the exponential phase (1.07 to 2.25 pg cell^−1^ for OA and 1.15 to 2.63 pg cell^−1^ for DTX-1) than in the stationary phase (0.14 to 0.73 pg cell^−1^ for OA and 0.25 to 1.25 pg cell^−1^ for DTX-1) when cells were grown on nitrate (Figure 11). An opposite trend was observed for experiments with ammonia as LST concentrations were slightly lower in the exponential phase (0.53–0.86 pg cell^−1^ for OA and 1.31–1.80 pg cell^−1^ for DTX-1) than in the stationary phase (1.83–6.11 pg cell^−1^ for OA and 3.1–4.4 pg cell^−1^ for DTX-1) (Figure 11). The growth rate did not differ in function of nitrate concentrations (Figure 12A). Cells grown with ammonia as the nitrogen source showed a decrease in the growth rate when the concentration of this nitrogen form increased with 0.18 and 0.17 d^−1^ at 441 and 882 µM, respectively. Cell length and width increased with increasing ammonia concentrations (Table 2). *P. lima* did not grow with an ammonia concentration of 1764 µM.

## 3. Discussion 

### 3.1. Ostreopsis *sp. 9*

*Ostreopsis* frequently co-occurs with other BHAB species of the genera *Prorocentrum* Ehrenberg and *Coolia* Meunier [8,32,109]. Genetically, the sequence of *Ostreopsis* sp. 9 OSCM17 strain from the Strait of Gibraltar closely aligns with strains named *Ostreopsis* cf. *siamensis* from the Mediterranean, Atlantic and Australia. The presence of this species in the Strait of Gibraltar is consistent with previous finding in the Mediterranean Sea, where *O*. cf. *siamensis* often co-occurs with the toxic *O*. cf. *ovata* [25,52,59]. Following its first molecular identification in Spain and Italy [35], *O*. cf. *siamensis* was subsequently identified in locations including the north Aegean Sea [52], Minorca (Balearic Islands) [57], Tunisian waters [59] and Catalonia, Spain [110] (Figure 13). Ibghi [34] reported the presence of *Ostreopsis* sp. in the Strait of Gibraltar for the first time, based on a field study conducted between 2019 and 2020. These authors recorded cell densities exceeding 2.7 × 10^5^ cells g^−1^ fresh weight of macrophyte, though specific species identification was not provided. In this study, we present the first formal identification of *Ostreopsis* sp. 9 in the Strait of Gibraltar on the basis of molecular ribotyping. In the Atlantic Ocean, *O*. cf. *siamensis* has been reported in various regions, including Brazil, Azores, Morocco, Portugal and, most recently, the Basque coast (France) [35,43,51,54,111,112,113] (Figure 13 and Figure 14, Table 3). Furthermore, this species has been identified in New Zealand and Australia [114] (Figure 14) and genetic analysis indicated only minor variations between populations from these regions [114]. Species of *Ostreopsis* are widely distributed and are predominantly observed between 35° N and 34° S latitude in tropical and subtropical regions. There has been an uptick in their presence in warm temperate marine ecosystems in recent years [20,115,116]. The first species of this genus to be described is *Ostreopsis siamensis* Schmidt, identified in the Gulf of Thailand (formerly known as the Gulf of Siam) [117]. Recent studies have determined it to be endemic to tropical areas such as the Pacific Ocean (Ryukyu Islands, French Polynesia), the Gulf of Thailand and the Caribbean Sea [46,47,49,118]. However, another species, though genetically distinct and not truly related to *O*. *siamensis*, has been ambiguously named ‘*O.* cf. *siamensis*’. To prevent confusion, some recent studies propose calling this species *Ostreopsis* sp. 9 [46,49]. This later species has been identified in New Zealand [55,56], Malaysia [119] and Australia’s Coral Sea, including Heron and Lady Elliot Islands, Hoffmans Rocks, and the Tasmanian Sea [114,120]. Nevertheless, *O*. cf. *siamensis* seems to be the most widely distributed species globally (Figure 14, Table 3).

Historically, the classification of *Ostreopsis* species hinged on morphometric studies, with plate shapes of the thecae being analyzed under fluorescence microscopy and scanning electron microscopy. However, these morphological identifications often remain inconclusive. As of now, no morphological differences have been highlighted between Atlantic and Mediterranean strains of *O.* cf. *siamensis*. Cells from the Atlantic are akin to their Mediterranean counterparts, both being teardrop-shaped with a golden–brown chloroplast that fills nearly the entire cell, barring the ventral region [62]. The cell sizes observed in our study align with previous research. Specifically, the measurements are consistent with findings for strains isolated from the Mediterranean Sea as reported by Penna et al. [35] (length: 80–75 µm; width: 34–56 µm), Laza-Martinez et al. [57] (length: 51–67 µm; width: 33–56 µm) and Ciminiello et al. [62] (length: 50–62 µm; width: 41–51 µm). Additionally, they are similar to observations in New Zealand by Rhodes et al. [56] (length: 52–68 µm; width: 44–55 µm). Morphologically, cells of the OSCM17 strain resemble those from other studies. Notably, the thecal plate pattern offers little in the way of differentiation among *Ostreopsis* species, given their strikingly similar designs and inherent plasticity [35,46,115].

In our laboratory conditions, the OSCM17 strain’s maximum cell density was achieved after 28 days, reaching 8.7 × 10^3^ cells mL^−1^. The growth rate of our strain was 0.26 d^−1^. This observed rate was below 0.5 d^−1^ which was reported for *O*. cf. *siamensis* from Australia by Tanimoto et al. [124]. While numerous studies in the Mediterranean were carried out on *O*. cf. *ovata*, limited research concerned *O*. cf. *siamensis*. However, the growth rate we observed for OSCM17 was in line with rates found for *O*. cf. *ovata* in various marine regions [125,126,127]. Factors such as nutrient concentrations, temperature, salinity and the physiology of different populations could impact the growth rate of *Ostreopsis* species [8,124,128]. 

LC–MS/MS analysis of OSCM17-strain cells did not detect any known toxins, namely PLTXs or ovatoxins. Paz et al. [64] and Ciminiello et al. [62] found that Mediterranean strains of *O*. cf. *siamensis* are not toxicogenic. During *Ostreopsis* bloom periods in the Strait of Gibraltar from 2019 to 2020, no toxins were found in the collected samples [34]. Furthermore, the *O*. cf. *siamensis* strain from the Atlantic coast does not produce toxins of the PLTX group either [43,113]. However, Verma et al. [61] showed that PLTX-like compounds were produced by a strain of *O.* cf. *siamensis* from Australian waters and the LD_50_ of the cell extract by intraperitoneal injection in mice was 25.1 mg kg^−1^. Further studies have to be carried out on the cytotoxicity of *Ostreopsis* sp. 9 OSM17 strain and on its toxicity on mice and crustacean models such as *Artemia* and copepods. 

### 3.2. Prorocentrum lima

#### 3.2.1. Distribution and Growth Characteristics

*Prorocentrum lima* PLCM17 strain isolated from the Strait of Gibraltar exhibits morphological characteristics consistent with those reported for *P. lima* from other marine systems [19]. A molecular phylogenetic analysis of PLCM 17 confirmed its identity as *P. lima* (Figure 5). This species was first described in the Gulf of Sorrento, Italy, in the Mediterranean Sea, by Ehrenberg in 1860. Since then, it has been reported worldwide and recognized as a cosmopolitan species. It thrives on a plethora of benthic substrates such as macrophytes, rocks, sand, angiosperms, coral reefs and marine invertebrates [129]. Despite observations of similar morphology across different populations, recent molecular studies have revealed that *P. lima* represents cryptic diversity, leading to its classification as a species complex [94,129,130]. Some species have been distinguished from *P. lima* based on minor morphological differences combined with distinct genotypic profiles, such as *Prorocentrum caipirignum* [79] and *Prorocentrum porosum* [131]. Our results indicate that the PLCM17 strain shares a close genetic relation with other Mediterranean strains, including Sorrento-1 from the type locality. *P. lima* is prevalent in the Mediterranean Sea, with reports from the North Aegean Sea in Greece [52], the French Mediterranean coast [29,87], the Gulf of Tunis [21,32], Bizerte Lagoon in north Tunisia [8] and Genoa at Quarto de Mille in Italy [132]. In the Strait of Gibraltar, blooms of *P. lima* were documented from May 2019 to November 2020. The highest observed cell densities reached 8.2 × 10^3^ cells g^−1^ of fresh weight of macrophytes and 10^3^ cells L^−1^ in the water column [34]. Morphological measurements of the PLCM17 strain were consistent with sizes reported by Nagahama et al. [129] for various strains, with no noticeable alterations throughout its growth. The variation in cell shape, defined by the length-to-width ratio (R), was recorded as 1.38 during the exponential phase and 1.36 during the stationary phase. No significant size variations were detected between these growth phases. Our findings are congruent with those observed in strains of *P. lima* in Tunisian waters [8,133], Greek coastal regions [15] and the northern South China Sea [94].

For the PLCM17 strain, we observed an initial growth phase lasting 38 days, followed by a nearly stagnant phase of cell concentration that lasted for 70 days and, thereafter, an exponential growth phase spanning 25 days. A similar extended exponential growth period was previously documented by Pan et al. [134] and Ben-Gharbia et al. [8]. The PLCM17 strain reached its maximum cell density of 49.5 × 10^3^ cells L^−1^ on the 145th day of culture, with a maximum growth rate of 0.21 d^−1^. When compared to other strains of *P. lima*, the growth rate of PLCM17 falls within the range of growth rates observed in various marine systems worldwide [8,134]. For instance, strains isolated from Heron Island in Australia exhibited a growth rate of 0.35 d^−1^ [135]. Furthermore, Wang et al. [136] noted that *P. lima* cell density reached 28.0 × 10^3^ cells L^−1^ after just 35 days of culture. Similarly, the PLBZT14 strain of *P. lima*, isolated from Bizerte lagoon (Tunisian Mediterranean waters) achieved a cell density of 32.10^3^ cells mL^−1^ after 60 days with a growth rate of 0.33 [8]. Due to variations in the inoculum concentrations and its physiological state employed across the laboratory experiments conducted in other studies, comparing growth kinetics of the investigated dinoflagellate remains challenging. It is advisable to primarily consider the maximum specific growth rate and cell yield of each alga. The growth rate we measured for the PLCM17 strain ranged from 0.17 to 0.22 at a temperature of 24 °C for the tested concentrations of ammonia and nitrate. These rates are slightly higher that those reported for some Atlantic strains of this dinoflagellate which exhibited growth rates ranging from 0.06 to 0.14 d^−1^ when cultivated at temperatures between 17 °C and 20 °C [69,137]. Contrastingly, some of the highest growth rates for *P. lima* have been reported in other locations of the Atlantic Ocean. Strains from Knight Key, Florida, that developed at a temperature of approximately 27 °C exhibited a growth rate of 0.75 d^−1^ [78]. Similarly, a strain isolated from the Portuguese coast showed a growth rate of 0.47 d^−1^ when cultivated at 19 °C [85]. In the present study, we observed an important decrease in PLCM17 growth rate at temperatures of 15 °C (0.08 d^−1^) and 29 °C (0.06 d^−1^) which suggests that temperature is a key element in the population dynamics of this species. However, other environmental factors such as nutrients, salinity and light conditions could influence the growth of *P. lima* and, in turn, its abundance in the marine ecosystem [136].

#### 3.2.2. Toxin Content

LSTs are responsible for widespread and recurrent LSP poisoning. In this study, both OA and DTX-1 were identified in the PLCM17 strain from the Strait of Gibraltar. When cells were cultivated at 24 °C, salinity of 36 and irradiance of 90 µmol photons m^−2^ s^−1^ on L1 medium, the toxin production for cells harvested on the 34th day (Figure 7B) was lower, with values of OA at 6.0 pg cell^−1^ and DTX-1 at 5.2 pg cell^−1^. The toxin concentration increased by the 145th day, with OA reaching 27.0 pg cell^−1^ and DTX-1 at 25.3 pg cell^−1^. In contrast, Ben-Gharbia et al. [8] found that for the Tunisian PLBZT14 strain of *P. lima*, LSTs peaked on the 12th day, as opposed to those recorded on the 60th day of cultivation. In terms of LST amounts, our results on toxin concentrations are close with those of Ben-Gharbia et al. [8] who showed that OA reached 28.3 pg cell^−1^. In the Pacific waters of Japan, the MIO12P strain of *P. lima* showed concentrations of OA and DTX-1 of 28.5 pg cell^−1^ and 23.6 pg cell^−1^, respectively [138]. Another strain from Okinawa, Japan, exhibited an OA concentration of 26.6 pg cell^−1^ and DTX-1 at 13.0 pg cell^−1^ [91]. The Japanese strain OMO29P showed concentrations of OA of 39.4 pg cell^−1^ and DTX-1 of 23.6 pg cell^−1^ when cells were cultivated at 25 °C [138]. In the Atlantic Ocean, elevated LST values for *P. lima* were reported. In Brazilian waters, OA was at 45.6 pg cell^−1^ [79], and in Portugal, OA and DTX-1 were recorded at 41.0 pg cell^−1^ and 12.0 pg cell^−1^, respectively [85]. OA in *P. lima* cells from UK waters was at 17.1 pg cell^−1^ and DTX-1 was at 11.3 pg cell^−1^ [69]. The most substantial concentrations of OA and DTX-1 were observed in natural cell samples from Strait of Gibraltar waters in Morocco, with OA ranging between 102.6 and 192.4 pg cell^−1^ and DTX-1 between 93.7 and 125.0 pg cell^−1^ for *P. lima* [34].

#### 3.2.3. Effect of Temperature on Toxin Content of *P. lima*

It has been demonstrated using controlled laboratory experiments that there is a variability in specific growth rates and OA and DTX-1 toxin content between *P. lima* strains [135,137]. The autoecology of *Prorocentrum* species remains poorly known [139]. However, previously conducted ecophysiological studies showed that *P. lima* was able to grow over a wide range of seawater temperatures (5–30 °C) [8,140]. Studies showed an efficient temperature adaptability of *P. lima* strains, explaining their worldwide distribution [8]. Lopez-Rosales et al. [141] suggested that reduced concentrations of intracellular LSTs in *Prorocentrm belizeanum* at the optimum growth temperature means that OA/DTX-1 toxins are secondary metabolites uncoupled from cell growth. Aquino-Cruz et al. [142] showed that the highest lipophilic toxicity in *P. lima* was found during the stationary growth phase at low (10–15 °C) and elevated (30 °C) temperatures. It has been shown that temperature could affect metabolic functions of *Prorocentrum* species such as growth, enzymatic activity and even DSP toxin production [140,141]. Hence, Wang et al. [136] demonstrated that a strain of *P. lima* increased several times its intracellular OA/DTX-1 toxin levels when cultured at low (15 °C) or high (30 °C) temperatures, disassociated from its optimum growth temperature [136]. Our results showed that *P. lima* PLM17 strain exhibited the highest OA/DTX-1 cellular contents at 24 °C, a temperature higher than that observed in situ during the summer season in the Strait of Gibraltar which did not exceed 21 °C [34]. These results suggest that global warming in the southern part of the Strait of Gibraltar may lead to higher growth rates and enhanced LST cellular toxin levels in *P. lima*. Our data showed that temperature seems to influence the LST production of *P. lima* as was shown for other dinoflagellate species. During the exponential phase, the maximum values were observed at 15 °C (11.63 pg cell^−1^ for OA and 15.27 pg cell^−1^ for DTX-1). In contrast, during the stationary phase, the maximum LST concentrations were registered at 24 °C (26.97 pg cell^−1^ for OA, and 25.27 pg cell^−1^ for DTX-1). One can suppose that the decrease in growth rate at 15 and 29 °C resulted in the increase in the proportion of cells which arrested their division and, in turn, increased the production of LSTs. Cells become larger with more toxins inside. 

#### 3.2.4. Effect of Nitrogen Source on LST Content of *P. lima*

Nitrogen is essential in the metabolism of phytoplankton as it is a key element in protein synthesis and the photosynthesis metabolism [143]. Many dinoflagellates exhibit mixotrophic behavior in their feeding strategy including osmotrophy and phagotrophy [144,145,146]. It has been demonstrated that some *Prorocentrum* species can use ammonia, nitrate, urea and amino acids for their nitrogen supply which suggests that species of this genera use osmotrophy for their nitrogen supply [134,147]. McLachlan et al. [143] showed that nitrogen concentration was positively correlated to the maximum cell density and growth rate of *P. lima*. Additionally, a substantial increase in cellular OA and DTXs levels was observed in *P. lima* grown in a nitrogen-limiting culture with less than 882 µM nitrate and, more generally, when *P. lima* cells were exposed to nutrients stress [111,136,137,140]. In our experiments with nitrate and ammonia as nitrogen sources, *P. lima* was grown at a temperature of 24 °C. In the presence of nitrate, OA and DTX-1 levels decreased when the concentration of this nitrogen was 441 µM and particularly during the stationary phase. This suggests that nitrate was consumed and the remaining nitrogen was not sufficient to sustain high level of LST production. Other studies showed that the highest OA and DTX-1 toxin production was observed during the stationary phase of growth of *P. lima* [148]. This trend was observed in our study but not when cells were cultured with ammonia as a nitrogen source. The concentration of phosphorus is important as the depletion of this nutrient leads to higher intracellular content of LSTs in *P. lima* [140,149]. It has been shown that the presence of ammonia does not favor toxin production in *Prorocentrum* species [134,137,147,148]. Our results showed that in the presence of ammonia, the concentrations of OA and DTX-1 in *P. lima* cells are higher in the stationary phase of growth in comparison to those measured in the exponential phase. In our study, cells cultivated under ammonia showed a decrease in the growth rate when the concentration of this nitrogen form increased with measured values of 0.18 and 0.17 d^−1^ at 441 and 882 µM, respectively. *P. lima* arrested its growth when the given ammonia concentration was 1764 µM. Our results corroborated those of Collos and Harrison [150] who showed that ammonia could be toxic to some dinoflagellate species at concentrations higher than 1200 µM. Given that our cultures were not acclimated to the tested concentrations of different nitrogen sources, the results should be considered cautiously. In further studies, it would be interesting to conduct a gradual adaptation of *P. lima* to increasing concentrations of various nitrogen forms, particularly ammonia, along with an investigation of the mechanisms involving genes encoding enzymes implicated in the assimilation and transformation of this molecule. The same acclimation strategy has to be considered in further laboratory experiments when investigating the effect of temperature and salinity on *P. lima* growth and toxicity.

### 3.3. Coolia monotis

Originally described as from the North Sea in Nieuwpoort, Belgium, by Meunier [96], the genus *Coolia* has since been recognized as being more diverse in tropical regions, particularly in estuarine and coastal waters [151]. *C. monotis* is a typically temperate species but has shown adaptability, flourishing in both tropical and subtropical ecosystems [8]. As highlighted by Abdennadher et al. [152], *C. monotis* can be considered as a eurytherm species, appearing in temperatures ranging from 9 to 31 °C and salinities reaching up to 59 [152]. This dinoflagellate has been recorded in Pacific waters, notably on coral reefs [47], in locations such as Okinawa, Ishigaki Island and the coast of Motobu, Japan [90,153]. It has also been spotted in Platypus Bay, Queensland, Australia [98], in Northland, Ninety Mile Beach and Rangiputa, New Zealand [56], the South China Sea near Hainan Island [154], the Rarotongan lagoons in the Cook Islands [155], the coast of Vietnam [156], the fringing reef off Sampadi Island, Sarawak, Malaysia [157], and the fringing reef surrounding Rawa Island, Malaysia [158].

In the Atlantic Ocean, *C. monotis* has been reported as being from various locations. These include Great Britain [148,159], the Netherlands [160], the southeastern Bay of Biscay [57] and Spain, specifically from San Sebastian in the north, Vigo in the west–central region and Galé in the south [161]. Additionally, the species was found in Florida, USA [35,56], as well as in Canada [162]. Within Mediterranean waters, *C. monotis* was observed in the Ionian Sea in Taranto, Italy [35], and in the north Aegean Sea in Greece [52,97]. Over the years, there has been a concerted effort to study dinoflagellates in the southern Mediterranean, revealing the consistent presence of *C. monotis* along its coasts [19]. Notably, the majority of data from the southern Mediterranean come from the Tunisian coasts: Marsa Bay [21], Mahdia [163], Bizerte Bay [30], Oued Lafann Chebba [31], the Gulf of Tunis [32] and the Gulf of Gabes [152,164]. In the Mediterranean waters of Morocco, *C. monotis* has been detected in the water column and on a majority of macrophytes, such as *Asparagopsis armata*, *Plocamium coccineum*, *Dictyoa dichotoma*, *Halopteris scoparia* and *Cladostephus spongiosus*, with a peak cell density of 4.1 × 10^4^ cells g^−1^ FW of macrophytes [34].

The morphology of the CMON15 strain corresponds to Meunier’s description [96]. The average cell length and width ranged from 34.92 to 34.99 µm and from 32.28 to 32.53 µm, respectively. Regarding the cell shape variation, the length/width ratio (R) was 1.08 during the exponential phase and 1.07 during the stationary phase. These findings are consistent with data reported on some strains, isolated from the southern Mediterranean of Tunisia [164]. For other strains of *C. monotis* isolated from the southern Mediterranean of Tunisia, the length ranged from 23.3 to 39.0 µm, and the width varied from 30.6 to 40.1 µm [164]. Globally, *C. monotis* is characterized by wide variation in its length and width, which vary between 23 and 55 µm and between 22.9 and 40.9 µm, respectively [8,97,151,162,164,165,166]. The maximum cell density for the Moroccan strain of *C. monotis*, CMON15, reached 10 × 10^3^ cells mL^−1^, observed after 28 days of cultivation, with a growth rate of 0.12 d^−1^. This finding corroborates previous works on *C. monotis*, which indicated growth rates of up to 0.6 d^−1^ [78,162,164]. The growth rate of *Ostreopsis* sp. 9 OSCM17 surpasses that of *P. lima* PLCM17 and *C. monotis* CMCN15, which could explain the dominance of *Ostreopsis* sp. 9 in the Strait of Gibraltar [the present study, 34]. A similar trend has been reported in the gulf of Bizerte, Tunisia, for *O*. cf. *ovata* [8].

Few studies have focused on the toxicity of *C. monotis* [164]. Cooliatoxin was first detected in a strain isolated from Platypus Bay, Queensland, Australia, in July 1988 which was initially identified as *C. monotis* [98]. However, subsequent research revealed that the species in question was *Coolia tropicalis* [167]. Other compounds, such as 44-methyl gambierone, were identified in strains of *C. tropicalis* [168]. In this study, the CMON15 strain from the Moroccan Strait of Gibraltar does not produce any known toxic compounds, as LC–MS/MS analyses did not detect peaks for cooliatoxin or yessotoxins. Ben-Gharbia et al. [8] identified a chromatographic peak at 5.6 min with a mass of *m*/*z* = 1061.768, closely resembling that of cooliatoxin (1061.5) in the *C. monotis* CMBZT14 strain isolated from the Mediterranean Sea which was not the case here for the CMON15 strain from the Strait of Gibraltar. No toxicity was detected in other strains from the Mediterranean and Atlantic regions [35,57,162,169].

## 4. Conclusions 

Compared to other benthic species, *Ostreopsis* sp. 9 (OSCM17) exhibits a high growth rate, which suggests that *Ostreopsis* species including toxic strains could adapt easily to environmental conditions in the Strait of Gibraltar. The influx of Atlantic waters into the Mediterranean via the Strait of Gibraltar may facilitate the introduction of other BHAB species into the region from adjacent Atlantic waters including toxic species such as *Ostreopsis ovata.* The presence of a major port, Tangier Med, with significant commercial boat traffic to the Strait of Gibraltar, heightens the risk of ballast water discharges, even those coming from tropical regions. This can lead to the introduction of emergent BHAB species, which could establish locally. 

DTX-1 and OA were detected in *P. lima* (PLCL17) isolated from the southern part of the Strait of Gibraltar, confirming the responsibility of *P. lima* in LST intoxication together with *Dinophysis* in southern Mediterranean waters [34,170]. The toxin content of this species varied depending on the growth phase. Nitrate and ammonia had an influence on the toxin content of *P. lima*, although no distinct trend was observed. *P. lima* arrested its growth when ammonia was given at a concentration of 1764 µM, corroborating the review paper by Collos and Harrison [150] on the toxicity of this nitrogen source on different phytoplankton taxa. *P. lima* grows well at 24 °C which is higher than the maximum summer temperature (21 °C) measured in coastal Strait of Gibraltar waters [34]. Given the ongoing warming of Mediterranean waters, alongside the growth of aquaculture and tourism activities, national monitoring programs should prioritize BHAB species and emerging toxins. This is especially relevant given the current focus on PSTs, ASTs and LSTs associated with planktonic toxic species of the *Alexandrium*, *Gymnodinium*, *Dinophysis* and *Pseudo-nitzschia* genera [105,106].

## 5. Material and Methods

### 5.1. Sampling Sites

*Ostreopsis* sp. 9 and *Prorocentrum lima* were collected from macrophytes at Cap Malabata (Strait of Gibraltar, Moroccan Mediterranean, 35°48′43.34″ N–5°45′1.07″ W) in June 2017. *Coolia monotis* was sampled in June 2015 from macrophytes found in Oued Negro (Western Moroccan Mediterranean coast (35°45′13.90″ N–5°18′53.39″ W) (Figure 15). The Western Moroccan Mediterranean coast is distinguished by its high urbanization level and is home to the major port of Tangier. This port experiences significant maritime activity, having hosted 10,902 ships in 2021 alone, which included 929 mega-ships measuring over 290 m in length (https://hbs.unctad.org/maritime-transport-indicators/, accessed on 1 April 2021).

### 5.2. Isolation and Culture Conditions

After collecting the macrophytes, BHAB cells were dislodged from the macroalgal surfaces, filtered through a 10 µm net, rinsed with seawater and then placed in 500 mL bottles filled with ENSW medium (Enriched Natural Sea Water, according to Harrison) [171]. The isolation of vegetative cells was accomplished using the capillary pipette method, observed under an inverted photonic microscope. *Ostreopsis* sp. 9 (designated OSCM17, from *Ostreopsis* sp. 9, Cap Malabata, 2017), *C. monotis* (CMON15, from *C. monotis*, Oued Negro, 2015) and *P. lima* (PLCM17, from *P. lima*, Cap Malabata, 2017). These cultures were maintained in two different media. OSCM17 and CMON15 were cultivated in ENSW medium, while PLCM17 was nurtured in L1 medium [172]. All strains were kept under stable conditions: 24 °C, a salinity of 36 and an irradiance of 90 µmol photons m^−2^ s^−1^, following a 12 h:12 h dark/light cycle.

### 5.3. Morphology and Identification

A DMi8 manual inverted microscope by Leica Microsystems (Wetzla, Germany) was employed for the morphometric measurements of the three BHAB species cultivated under optimal conditions (24 °C, salinity of 36 and 90 µmol photons m^−2^ s^−1^). During both the exponential and stationary growth phases, a minimum of 30 cells were collected and preserved with Lugol’s solution. The cell length and width were determined at 400× magnification using a camera coupled with ProgRess Capture 2.9.0.1 software. After staining with calcofluor, the cells were examined under a Leica epifluorescent microscope (Leica Microsystems model CMS GmbH, Germany) to identify the species based on the tabulation of thecal plates.

### 5.4. Molecular Analysis and Phylogeny

#### 5.4.1. DNA Extraction and PCR

The three strains were extracted with the PCRBIO Rapid Extract PCR Kit (PCR Biosystems Ltd.) which combines extraction and PCR. In a 1.5 mL tube, 1 mL of culture was taken and centrifuged for 3 min at 17,900× *g*. The supernatant was discarded to retain only the pellet. Then, the manufacturer’s instructions were followed, except for the step dilution, where 190 µL of PCR-grade dH_2_O was added instead of 900 µL. Targeting the LSU rDNA gene partial, the pair of primers D1R (ACCCGCTGAATTTAAGCATA) [173] and D3B (TCGGAGGGAACCAGCTACTA) [174] were used with a Tm of 56 °C. In addition, the D8 region was amplified for strains OSCM17 and CMON15 with primers FD8 (GGATTGGCTCTGAGGGTTGGG) [175] and RB (GATAGGAAGAGCCGACATCGA) [175] with a Tm of 55 °C.

#### 5.4.2. Amplification and Sequencing

PCR-amplified product was visualized on agarose gel after electrophoresis and the positive samples were purified using the ExoSAP-IT PCR Product Cleanup reagent (Affymetrix, Cleveland, OH, USA). The Big Dye Terminator v3.1 Cycle Sequencing Kit (Applied Biosystems, Tokyo, Japan) was used for sequencing of the amplicon generated. Primers and excess dye-labeled nucleotides were first removed using the Big Dye X-terminator purification kit (Applied Biosystems, Foster City, CA, USA). Sequencing products were run on an ABI PRISM 3130 Genetic Analyzer (Applied Biosystems). 

#### 5.4.3. Phylogeny

For the phylogenetic analyses, the sequences of the three BHAB strains were aligned along with other related sequences in two separate datasets. Since *Ostreopsis* and *Coolia* are phylogenetically closely related genera [176], we conducted a single phylogenetic analysis that included both genera. This analysis was based on the concatenated D1–D3 and D8–D10 domains of the large ribosomal subunit. To accomplish this, we incorporated sequences from strains OSCM17 and CMON15, along with 68 sequences from *Ostreopsis* (48 sequences), *Coolia* (15 sequences) and *Alexandrium* (5 sequences, serving as the outgroup), using Boisnoir’s analysis [49] as a foundation. The two regions of the LSU were initially aligned separately using the MAFFT algorithm with the q-ins-i option in Version 7 [108]. Subsequently, we concatenated them using SeaView Version 4 software [177]. The final alignment encompassed 70 sequences and 2003 positions, including gaps. For *P. lima*, we utilized a matrix of 977 nt, which consisted of the D1–D3 domains of the LSU, along with 36 sequences. This dataset included the PLCM17 strain, 33 sequences of *Prorocentrum* and 2 sequences of *Scrippsiella* (serving as the outgroup), retrieved from GenBank. Sequences were aligned using MUSCLE software version 3.7 [178]. Both datasets underwent manual refinement and were subjected to two methods of phylogenetic reconstruction: maximum likelihood (ML) using PhyML version 3.0 software [179] and Bayesian inference (BI) using MrBayes version 3.1.2 [180]. To determine the most suitable model of substitutions, we initially employed jModeltest version 0.1.1 [181]. The general time-reversible model (GTR + I + G) was selected based on hierarchical likelihood ratio tests (hLRTs), Akaike Information Criterion 1 (AIC1), Akaike Information Criterion 2 (AIC2) and Bayesian information criterion (BIC) tests implemented in jModeltest. Bootstrap values, representing support for branches in the trees, were generated through 1000 iterations in the ML analysis. In Bayesian inference, four Markov chains were run simultaneously for 2 × 10^6^ generations with sampling occurring every 100 generations. Out of the 2 × 10^4^ trees obtained, the initial 2000 were discarded as part of the burn-in process, and a consensus tree was constructed using the remaining trees. Posterior probabilities, indicating the frequency of a node’s presence in the preserved trees, were calculated utilizing a coupled Monte Carlo Metropolis approach–Markov Chain (MCMC). Accession numbers for sequences are provided in brackets (with ‘Ø’ indicating the absence of one region). Branch support is indicated by bootstrap values (ML) and posterior probabilities (BI). Bootstrap values below 65 and posterior probabilities below 0.5 are represented with ‘-’. On the right, vertical bars delineate major clades corresponding to different species/genotypes.

### 5.5. Toxin Analysis 

During the exponential and stationary growth phases of the three BHAB cultures, 20 mL of each culture was collected for toxin analyses. The samples were centrifuged at 4000× *g* for 10 min and the pellet was stored at −20 °C. The pellets of the *C. monotis* cells were suspended in 500 µL of 90% methanol; pellets of *Ostreopsis* sp. 9 and *P. lima* cells were resuspended in 500 µL of 100% methanol. Suspended cells were ground with glass beads (125 mg) for 20 min in a grinder. After centrifugation at 10,000× *g* for 10 min, the supernatant was collected and filtered through a 0.2 µm filter for toxin analysis by liquid chromatography–tandem mass spectrometry (LC–MS/MS).

#### 5.5.1. LC–MS/MS Analysis of Lipophilic Toxins Produced by Strain PLCM17 

The sample analyses were conducted using a UFLC (model UFLC, Shimadzu) coupled to a triple-quadrupole mass spectrometer (4000 Qtrap, ABSciex) equipped with a turboV^®^ ESI source. Chromatographic separation was performed on a C18 Kinetex column (100 Å, 2.6 μm, 100 × 2.1 mm, Phenomenex) along with a C18 guard column (4 × 2.0 mm, 2.6 μm, Phenomenex). A binary mobile phase consisting of phase A (100% aqueous) and phase B (95% aqueous acetonitrile), both containing 2 mM ammonia formate and 50 mM formic acid, was employed. The flow rate was set at 0.4 mL min^−1^, and the injection volume was 5 μL. The column and sample temperatures were maintained at 40 °C and 4 °C, respectively. A gradient elution method was used, commencing with 20% B, increasing to 95% B over 8 min, holding for 3 min, then reducing to 20% B in 0.5 min and maintaining this level for 3 min to equilibrate the system. For quantification purposes, the mass spectrometer operated in the multiple reactions monitoring (MRM) acquisition mode, scanning two transitions for each toxin. Negative acquisition experiments were set up with the following source settings: curtain gas set at 20 psi, ion spray at −4500 V, temperature at 550 °C, gas 1 and 2 set at 40 and 55 psi, respectively, and an entrance potential of 13 V. These parameters were previously optimized using toxin standards. The mass spectrometer operated in MRM mode, analyzing two product ions per compound. For each toxin, the first transition, which was the most intense, was used for quantification. In ESI negative mode, the selected transition was as follows: [M–H]^−1^ ions: OA and DTX-2, *m*/*z* 803.4 > 255.1/113.1; DTX-1, *m*/*z* 817.5 > 254.9/112.9. Certified calibration solutions of OA, DTX-2 and DTX-1 were procured from the National Research Council Canada (NRCC, Halifax, NS, Canada).

#### 5.5.2. LC–MS/MS Analysis of Toxins Produced by Strain OSCM17

Liquid chromatography was carried out on a Poroshell 120 EC-C18 column (100 × 2.1 mm, 2.7 μm, Agilent) equipped with a guard column (5 × 2.1 mm, 2.7 μm, using the same stationary phase). An Ultra-Fast Liquid Chromatography system (Prominence UFLC-XR, Shimadzu) was employed for this purpose. Gradients of water (A) and acetonitrile (95%, B), both containing 0.2% acetic acid, were utilized at a flow rate of 0.2 mL min^−1^. The injection volume was 5 μL, and the column temperature was maintained at 25 °C. MS/MS analyses were conducted using a 4000 QTRAP (AB Sciex) in positive ion mode, employing multiple reactions monitoring (MRM) acquisition. UV detection at 220 nm, 233 nm, 263 nm and in the range of 220–360 nm was performed with a diode array detector (Prominence, SPD-M20A, Shimadzu). In total, two LC–MS/MS methods and one LC–UV–MS/MS method, as described in Chomérat [45], were employed to detect palytoxin, 42-OH-palytoxin, 12 ovatoxins (OVTX-a to -k), 3 ostreocins (OST-A, -B, -D and -E1), 3 mascarenotoxins (McTX-A to -C) and ostreotoxins-1 and -3. Quantification was carried out relative to a Palytoxin standard (Wako Chemicals GmbH, Germany) using a 6-point calibration curve.

#### 5.5.3. LC–MS/MS Analysis of Compounds Produced by Strain CMON15 

A screening of cooliatoxin, a monosulfated polyether toxin, was conducted using a system consisting of an ultra-high-performance liquid chromatography (UHPLC) system (UFLC, Shimadzu) coupled with a hybrid triple-quadrupole–linear ion-trap mass spectrometer (4000 QTRAP, Sciex) equipped with a TurboV source (ESI). Cooliatoxin screening was carried out following the method described by Ben-Gharbia et al. [8]. 

### 5.6. Growth Characteristics

Cell concentrations of the three BHAB species under study were monitored in triplicate cultures. To perform cell counting, a 5 mL sample was taken after homogenization in 250 mL Erlenmeyer flasks (one separate culture in each flask) at the same time of day every three or four days. This monitoring continued for 28 days for *Ostreopsis* sp. 9, 38 days for *C. monotis* and 155 days for *P. lima*. The collected samples were fixed and then placed (1 mL each) on a Sedgewick–Rafter slide. Cell counting was conducted using an inverted photonic microscope, following the method outlined by Guillard [182]. To calculate the maximum growth rate (expressed in day^−1^), a linear regression was performed over the entire exponential growth phase. This was achieved by applying the least square fit of a straight line to the data after logarithmic transformation. The formula used for calculating the growth rate was as follows: µm = (Ln(N_1_) − Ln(N_0_))/(T_1_ − T_0_), where µm represents the growth rate in units of day^−1^, and N_1_ and N_0_ denote the cell density at time T_1_ and T_0_, respectively, during the linear portion of the exponential growth phase.

### 5.7. Effect of Temperature and Nitrogen Sources (Ammonia and Nitrate) on Toxin Content of Prorocentrum lima

To test the effects of temperature on *P. lima* PLCM17 toxin content, 250 mL sterile flasks filled with 250 mL of culture medium were inoculated with cells taken during the exponential growth phase of each strain cultivated previously in the same conditions. The cultures were incubated at different temperatures (15, 20, 24 and 29 °C) in a calibrated incubator with an error of 1 °C. For this experiment alone, we gradually adapted *P. lima* to various tested temperatures (15, 20, 24, and 29 °C) by incrementally increasing or decreasing the initial temperature (24 °C) by 0.5 °C per day.

In order to test the effect of nitrogen on PLCM17 toxin content, the modified L1 and ENSW media containing various sources of nitrogen, nitrate (NaNO_3_) and ammonia (NH_4_Cl), were prepared with three concentrations (441, 882 and 1764 µM) of each nitrogen form. Concentrations were chosen by multiplying by two and dividing by two the concentration of the original L1 medium.

During the exponential and stationary growth phases of *P. lima*, 20 mL of each culture was collected for toxin analyses. The samples were centrifuged at 4000× *g* for 10 min and the pellet was stored at −20 °C. The pellets were dissolved with 1 mL of 90% methanol. In a mixing mill, the mixtures were ground with glass beads (0.25 g) for 30 min. The supernatant was collected and filtered through a 0.2 µm filter after centrifugation at 5000× *g* for 10 min, then it was injected in liquid chromatography coupled to tandem mass spectrometry (LC–MS/MS).

## Figures and Tables

**Figure 1 toxins-16-00049-f001:**
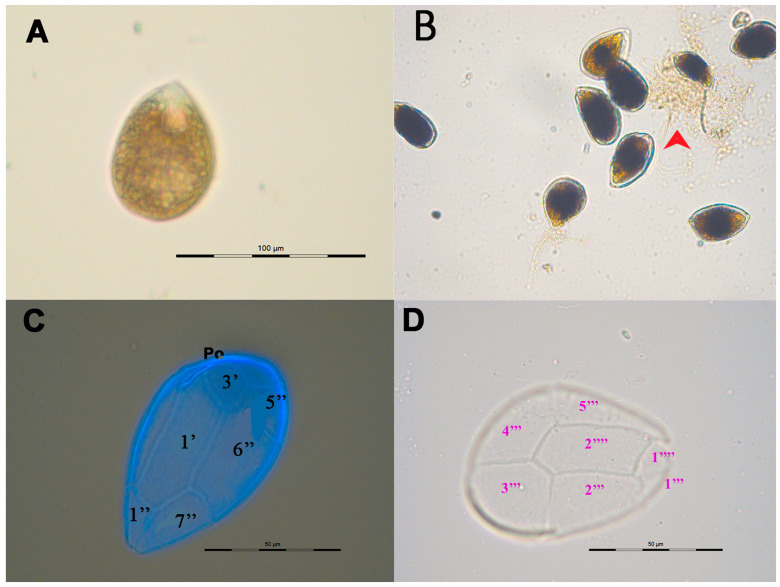
Vegetative cells of *Ostreopsis* sp. 9 OSCM17 strain, observed under light microscopy (**A**,**B**) and after calcofluor staining: (**A**) single cell; (**B**) cells embedded in mucus indicated by the arrow; (**C**) epithecal view and (**D**) hypothecal view with the tabulation. Po: pore plate.

**Figure 2 toxins-16-00049-f002:**
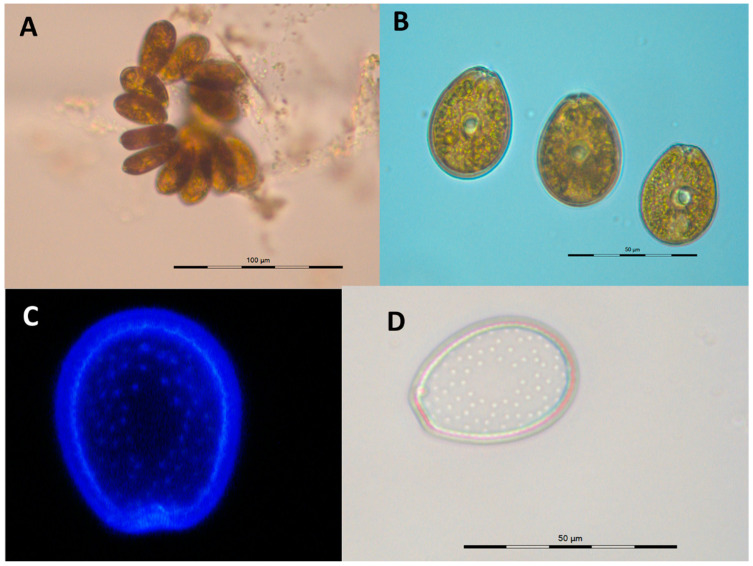
Cells of *Prorocentrum lima* PLCM17 strain observed under inverted photonic microscope (**A**,**B**), single cell after calcofluor staining. V-shaped right valve (**C**) and left valves (**D**).

**Figure 3 toxins-16-00049-f003:**
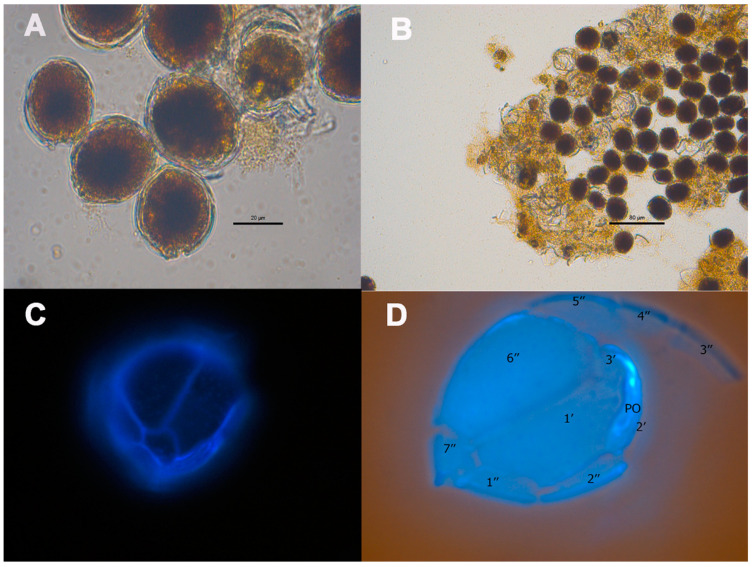
Vegetative cells of *Coolia monotis* CMON15 strain observed under inverted photonic microscope with mucus embedding cells (**A**,**B**). Cells stained with calcofluor (**C**,**D**), with hypothecal view and the corresponding tabulation. Po: pore plate.

**Figure 4 toxins-16-00049-f004:**
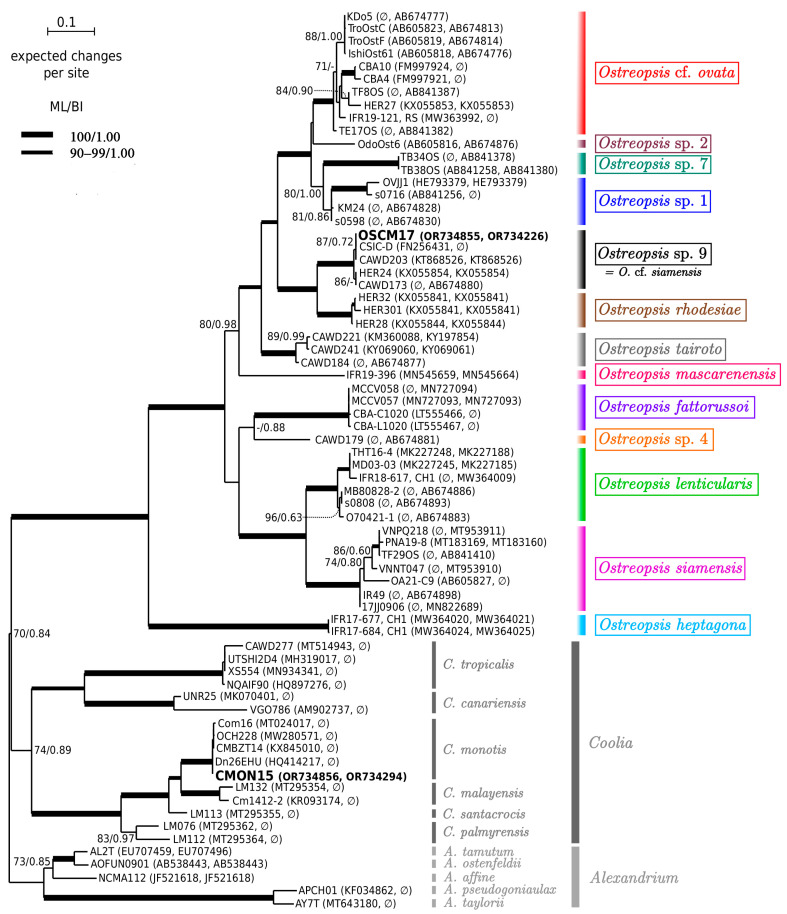
Phylogeny of *Ostreopsis* sp. 9 OSCM17 and *Coolia monotis* CMON15 based on 88 sequences of concatenated D1–D3 and D8–D10 domains of LSU rDNA using MAFFT version 7 (q-ins-i option) software [108].

**Figure 5 toxins-16-00049-f005:**
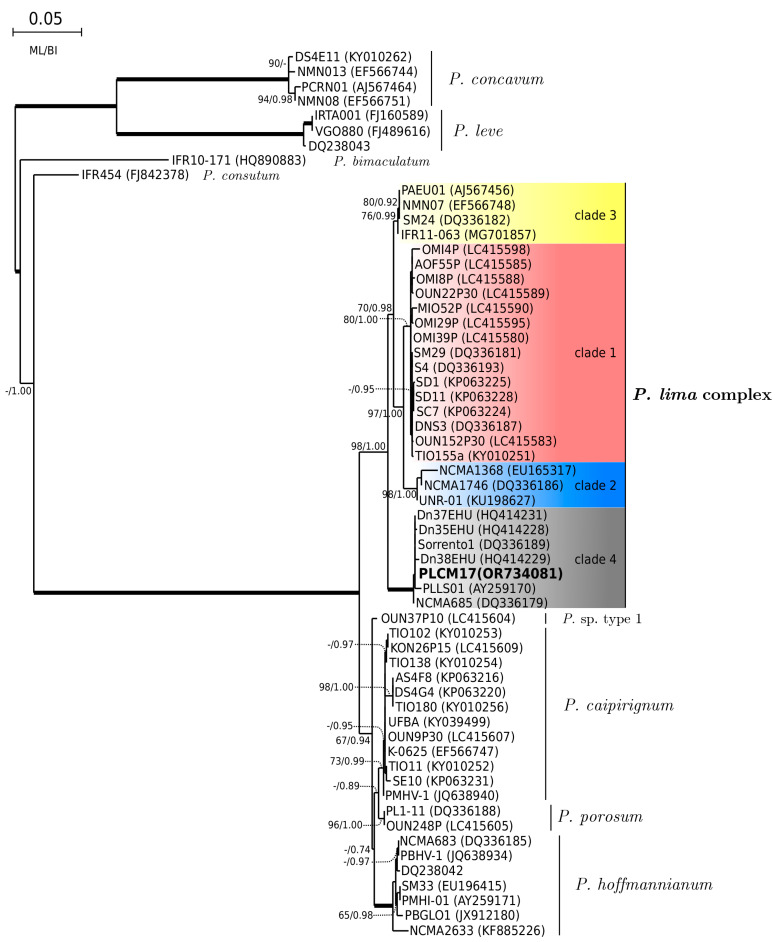
Phylogeny of *Prorocentrum lima* PLCM17 based on 88 sequences of concatenated D1–D3 domains of LSU rDNA using MAFFT version 7 (q-ins-i option) software [108].

**Figure 6 toxins-16-00049-f006:**
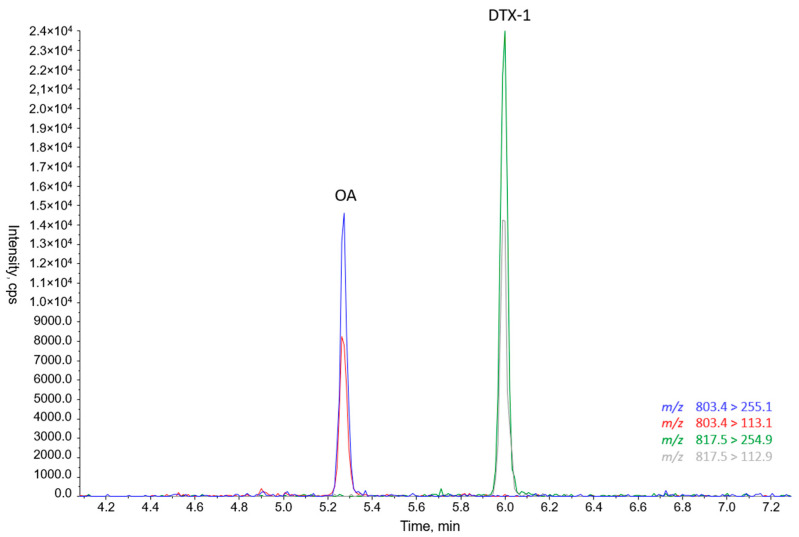
Chromatogram of LC–MS/MS analysis of PLCM17 *Prorocentrum lima* strain from Strait of Gibraltar, southern Mediterranean. OA and DTX-1 refer to okadaic acid and dinophysistoxin-1, respectively. The blue line corresponds to the *m*/*z* 803.4 > 255.1 transition, the red line to the 803.4 > 113.1 transition, the green line to the 817.5 > 254.9 transition, and the grey line to the 817.5 > 112.9 transition.

**Figure 7 toxins-16-00049-f007:**
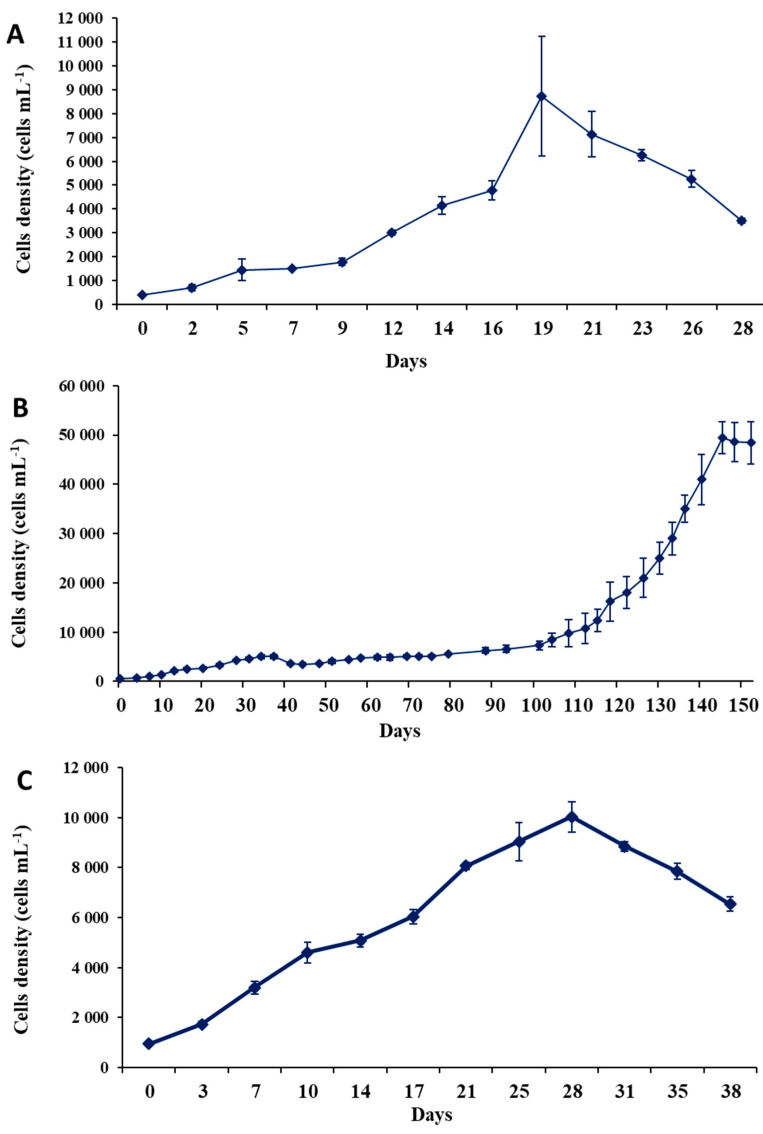
Growth patterns (±SD) of *Ostreopsis* sp. 9 OSCM17 (**A**), *Prorocentrum lima* PLCM17 (**B**) and *Coolia monotis* CMON15 strains (**C**) cultivated at temperature of 24 °C, salinity of 36 and irradiance of 90 µmol photons m^−2^ s^−1^ with 12 h:12 h dark/light cycle. SD: standard deviation.

**Figure 8 toxins-16-00049-f008:**
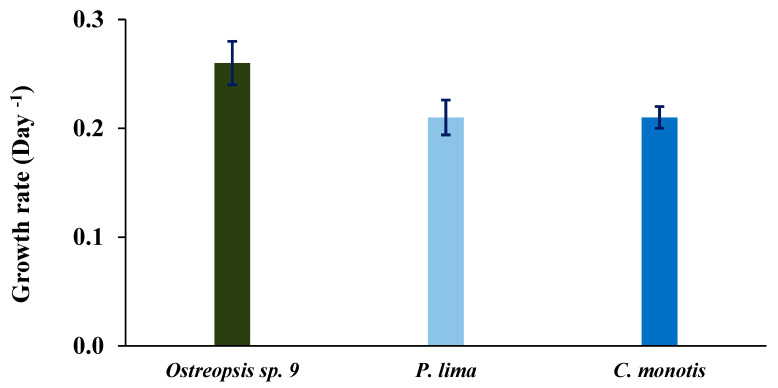
Growth rates (±SD) in day^−1^ of *Ostreopsis* sp. 9 OSCM17, *Prorocentrum lima* PLCM17 and *Coolia monotis* CMON15 cultivated at temperature of 24 °C, salinity of 36 and irradiance of 90 µmol photons m^−2^ s^−1^ with 12 h:12 h dark/light cycle. SD: standard deviation.

**Figure 9 toxins-16-00049-f009:**
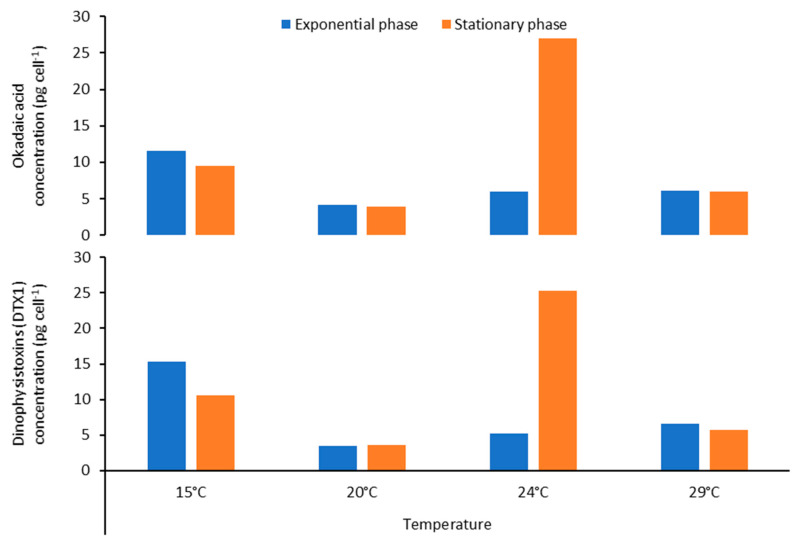
Okadaic acid and DTX-1 concentrations (in pg cell^−1^) measured during exponential and stationary growth phases of *Prorocentrum lima* PLCM17 strain cultivated at increasing temperatures: 15, 20, 24 and 29 °C.

**Figure 10 toxins-16-00049-f010:**
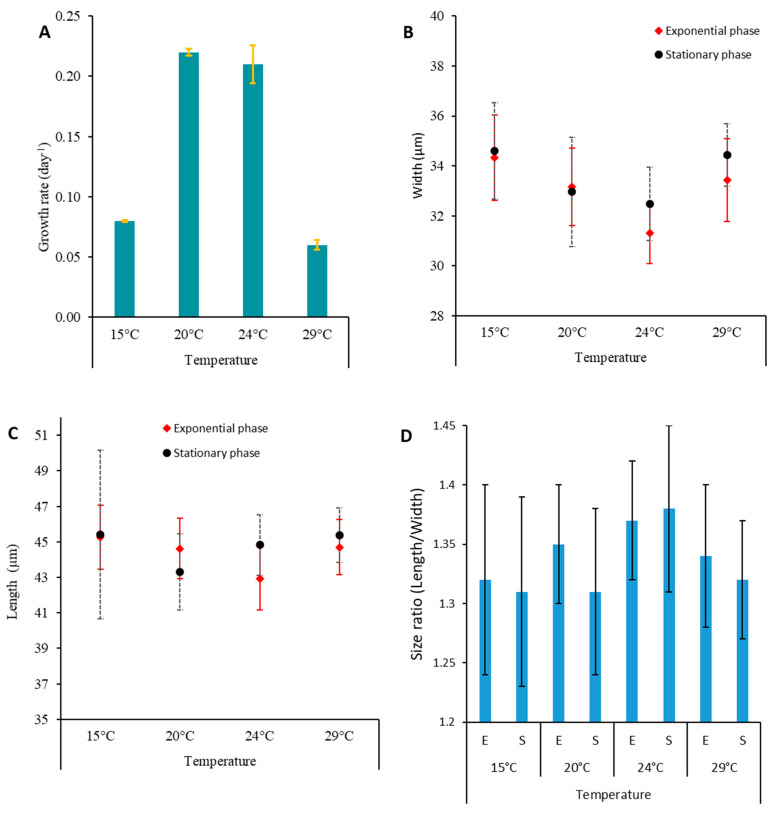
Growth rate (±SD) in d^−1^ (**A**), cell width (±SD) (**B**) and length (±SD) (**C**) in µm and length/width ratio (±SD) (**D**) measured at exponential and stationary growth phases of *Prorocentrum lima* PLCM17 strain cultivated at increasing temperatures (15, 20, 24 and 29 °C). The microalgae were cultivated at salinity of 36 and irradiance of 90 µmol photons m^−2^ s^−1^ with 12 h:12 h dark/light cycle. In (**D**), E and S mean exponential and stationary growth phases of the cultures, respectively. SD: standard deviation.

**Figure 11 toxins-16-00049-f011:**
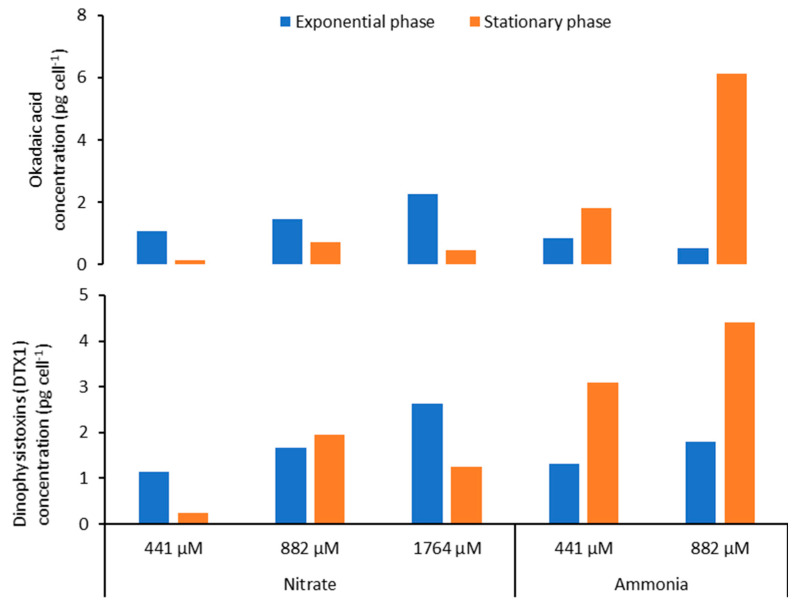
Okadaic acid and DTX-1 concentrations during exponential and stationary growth phases of *Prorocentrum lima* PLCM17 strain cultivated at different concentrations of nitrate and ammonia. Cultures were grown at a temperature of 24 °C, salinity of 36 and irradiance of 90 µmol photons m^−2^ s^−1^ with 12 h:12 h dark/light cycle.

**Figure 12 toxins-16-00049-f012:**
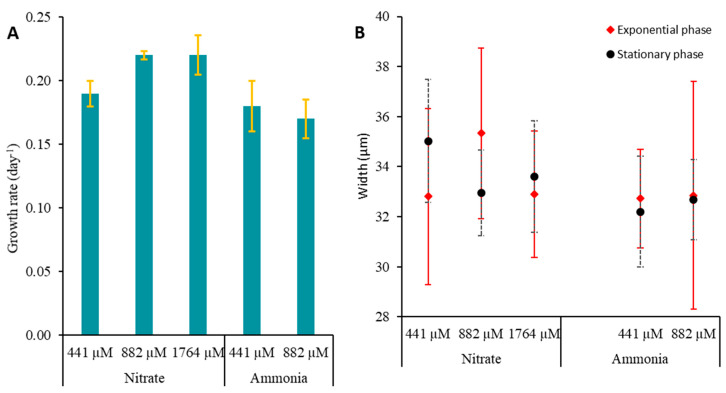
Growth rate (±SD) in d^−1^ (**A**), cell width (±SD) (**B**) and length (±SD) (**C**) in µm and length/width ratio (±SD) (**D**) measured at exponential and stationary growth phases of *Prorocentrum lima* PLCM17 strain cultivated at different concentrations of nitrate and ammonia. Cultures were carried out at a temperature of 24 °C, salinity of 36 and irradiance of 90 µmol photons m^−2^ s^−1^ with 12 h:12 h dark/light cycle. In (**D**), E and S refer to exponential and stationary growth phases, respectively. SD: standard deviation.

**Figure 13 toxins-16-00049-f013:**
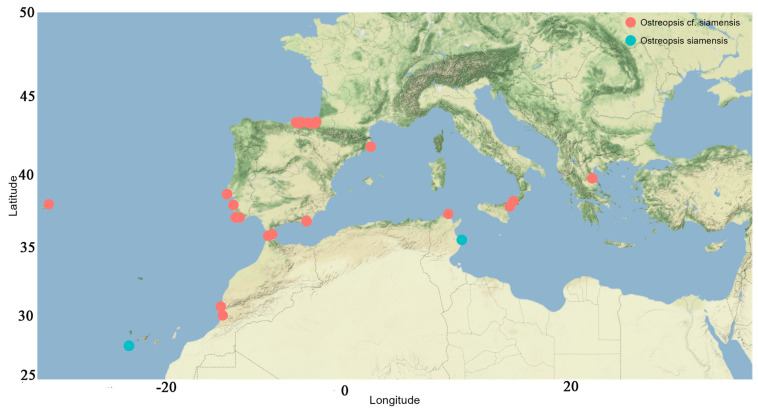
Geographic distribution of *Ostreopsis siamensis* and *Ostreopsis* cf. *siamensis* in the Mediterranean Sea and in some Atlantic areas near the Strait of Gibraltar.

**Figure 14 toxins-16-00049-f014:**
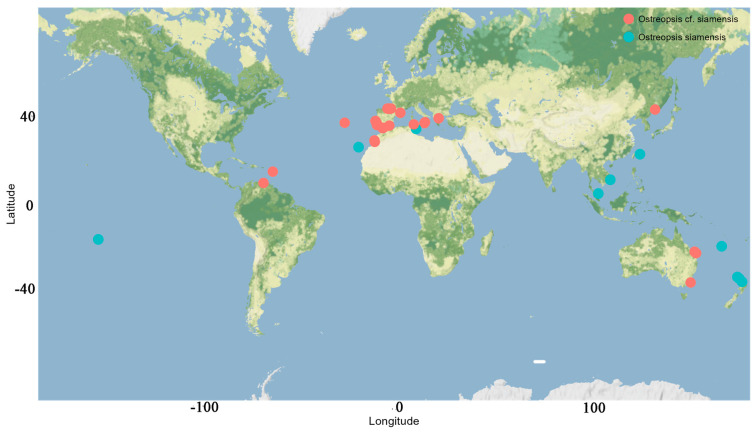
Global distribution of *Ostreopsis siamensis* and *Ostreopsis* cf. *siamensis*.

**Figure 15 toxins-16-00049-f015:**
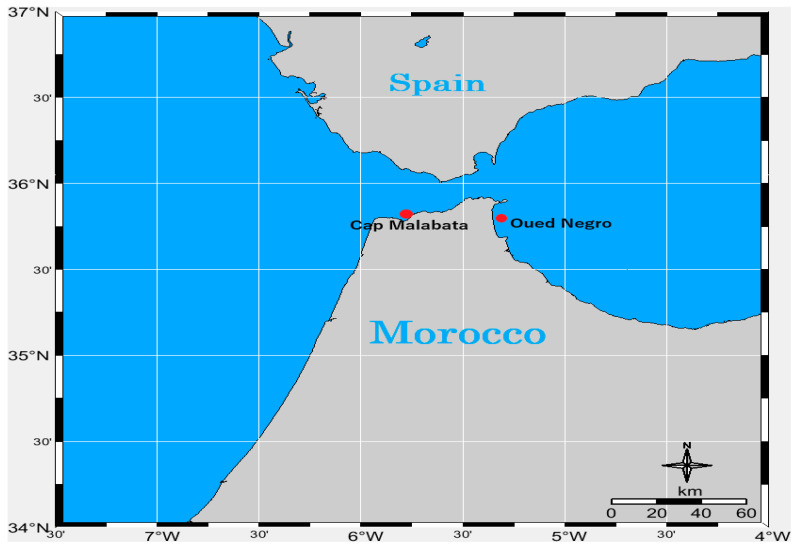
Map of the location of Cap Malabata and Oued Negro (southwestern Mediterranean, Morocco) where the benthic dinoflagellates were isolated (*Coolia monotis* CMON15 strain was isolated in Oued Negro in 2015 whereas *Ostreopsis* sp. 9 OSCM17 and *Prorocentrum lima* PLCM17 strains were isolated in Cap Malabata at 2017).

**Table 1 toxins-16-00049-t001:** Morphometric characteristics of *Ostreopsis* sp. 9 OSCM17, *Prorocentrum lima* PLCM17 and *Coolia monotis* CMON15 strains: mean, minimum, maximum values (µm) and standard deviation (SD) of the length and width of the cells harvested in both exponential and stationary growth phases (*n* ≥ 30). Cells were grown at 24 °C, salinity of 36 and an irradiance of 90 µmol phtons m^−2^ s^−1^ and with 12 h:12 h dark/light cycle.

		*Ostreopsis* sp. 9			*Prorocentrum lima*			*Coolia monotis*		
		Mean ± SD	Min	Max	Mean ± SD	Min	Max	Mean ± SD	Min	Max
Exponential phase	Length (µm)	48.81 ± 3.43	38.07	58.98	42.94 ± 1.79	37	47.83	34.92 ± 2.02	30.24	38.06
	Width (µm)	35.79 ± 3.56	28.67	43.67	31.3 ± 1.23	28.15	35.12	32.53 ± 2.09	28.21	36.64
Stationary phase	Length (µm)	53.24 ± 3.01	45.96	60.81	44.83 ± 1.70	40.43	48.29	34.99 ± 2.58	30.19	39.47
	Width (µm)	36.20 ± 2.39	30.02	44.57	32.46 ± 1.43	28.91	35.38	32.28 ± 2.36	27.94	38

**Table 2 toxins-16-00049-t002:** Okadaic acid (pg cell^−1^), DTX-1 (pg cell^−1^), mean length (µm), mean width (µm), size ratio (Length/Width) and maximum growth rate (day^−1^) of *Prorocentrum lima* PLCM17 strain cultivated in varying conditions of temperature (15 °C, 20 °C, 24 °C, 29 °C), nitrate (441 µM, 882 µM, 1764 µM) and ammonia (441 µM, 882 µM). For nitrate and ammonia experiments, *P. lima* was cultivated at 24 °C.

		Okadaic Acid		DTX-1		Length (µm)		Width (µm)		Size Ratio (L/W)		Maximum Growth Rate (d^−1^)
		Exp	Stat	Exp	Stat	Exp	Stat	Exp	Stat	Exp	Stat	
Temperature	15 °C	11.63	9.51	15.27	10.64	45.26	45.42	34.32	34.61	1.32	1.31	0.08
	20 °C	4.21	3.90	3.47	3.58	44.62	43.31	33.17	32.96	1.35	1.31	0.22
	24 °C	6.04	26.97	5.19	25.27	42.94	44.83	31.31	32.49	1.37	1.38	0.21
	29 °C	6.05	5.98	6.53	5.75	44.7	45.37	33.43	34.44	1.34	1.32	0.06
Nitrate	441 µM	1.07	0.14	1.15	0.25	41.42	42.67	32.81	35.03	1.26	1.22	0.19
	882 µM	1.48	0.73	1.66	1.95	42.91	42.53	35.33	32.94	1.21	1.29	0.21
	1764 µM	2.25	0.47	2.63	1.25	42.12	43.05	32.9	33.6	1.28	1.28	0.22
Ammonia	441 µM	0.86	1.83	1.31	3.10	40.5	40.46	32.73	32.2	1.24	1.26	0.18
	882 µM	0.53	6.11	1.80	4.40	39.84	43.59	32.85	32.69	1.21	1.33	0.17

**Table 3 toxins-16-00049-t003:** Summary for *Ostreopsis* cf. *siamensis*, *Ostreopsis* sp. 9 and *Ostreopsis siamensis* from various marine ecosystems. Culture medium, toxicity, growth rates and cells densities are specified when available.

	Latitude	Longitude	Year of Observation/Isolation	In Situ or Lab	Temp (°C)	Sal	Light µmol Photons m^−2^ s^−1^	Toxicity	Growth Rate (µ, d^−1^)	Maximum Cell Density FWM: Fresh Weight of Macrophyte	Ref.
Mediterranean water											
Greece											
*Ostreopsis* cf. *siamensis*	39.76141°	22.50722°	2003–2004	In situ	13.9–29.7	16.7–37.1	–	-	-	4.05 × 10^5^ cells g^−1^ FWM	[52]
	39.76141°	22.50722°	2003–2005	f/2 or K	19.0 ± 1	-	70	-	-		
Italy											
*Ostreopsis* cf. *siamensis*	37.85011°	15.30071°	2005–2009	f/2	23.1	-	100	Not toxic	-	-	[62]
	38.22121°	15.60987°	1999–2000	K, f/20, f/2	17 ± 1	-	100	-	-	-	[35]
France											
*Ostreopsis* cf. *siamensis*	43.40271°	−2.36951°	2018	In situ	14.0–25.9	24.8–37.0	-	-	-	10^5^ cells g^−1^ FWM	[113]
	43.39860°	−1.66280°	2010	f/4	20	-	80	-	-	-	[51]
Spain											
*Ostreopsis* cf. *siamensis*	41.85562°	3.14320°	2016	f/2	24 ± 2	36	110	-	-	-	[110]
	36.82037°	−2.44678°									
	36.81699°	−2.46840°	1999–2000	K, f/20, f/2	17 ± 1	-	100	-	-	-	[35]
	41.84566°	3.146161°									
Morocco											
*Ostreopsis* sp. 9	35.81203°	−5.75029°	2017	ENSW	24	36	90	Not toxic	0.18	8.75 × 10^3^ cells mL^−1^	This study
Tunisia											
*Ostreopsis siamensis*	35.50249°	11.09569°	2008–2009	In situ	-	-	-	-	-	≥5 × 10^2^ cells.100 g^−1^ FWM	[59]
*Ostreopsis* cf. *siamensis*	37.32533°	9.92102°	2008	In situ	-	-	-	-	-	3.75 × 10^4^ cells L^−1^	[59]
Atlantic Water											
France											
*Ostreopsis* sp. 9	43.44881°	−1.60637°	2021	L1	17	35	160	Not toxic	-	9.63 × 10^4^ cells mL^−1^	[43]
Guadeloupe France											
*Ostreopsis siamensis*	16.17203°	−61.36256°	2017–2018	In situ	-	-	-	-	-	-	[49]
Spain											
*Ostreopsis* cf. *siamensis*	43.41660°	−3.39960°	2010	f/4	20	-	80	-	-	-	[51]
	43.35250°	−3.07800°	2010	f/4	20	-	80	-	-	-	
	43.32150°	−1.9870°	2010–2011	f/4	20	-	80	-	-	-	
	43.44332°	−2.981080°	2007–2009	f/2	17–22	30–35	60	Toxic for *Artemia* nauplii	-	-	[57]
*Ostreopsis siamensis*	27.63750°	−17.98833°	2017	In situ	-	-	-	-	-	-	[44]
Portugal											
*Ostreopsis* cf. *siamensis*	38.00000°	−25.00000°	2009	In situ	20–24	-	-	-	-	10^5^ cells L^−1^	[121]
	37.95333°	−8.85883°	2008–2009	f/20-Si	19	35	40	-	-	-	[50]
	37.08150°	−8.66820°	2011	f/4	20	-	80	-	-	-	[51]
	38.70180°	−9.40840°	2010	f/4	20	-	80	-	-	-	
	37.07990°	−8.31560°	2010–2011	f/4	20	-	80	-	-	-	
	38.69356°	−9.41470°	2010–2011	f/20	20	33–34	-	-	-	-	[112]
	37.09126°	−8.66902°	2015–2016	f/20	20	33–34	-	-	-	-	
	37.95055°	−8.86471°	2005–2009	f/2	23.1	-	100	Not toxic	-	-	[62]
	38.69333°	−9.40927°	2005–2009	f/2	23.1	-	100	Not toxic	-	-	
Morocco											
*Ostreopsis* cf. *siamensis*	30.62479°	−9.95794°	2009	In situ	20–24	-	-	Toxic (Mouse Bioassay)	-	10^5^ cells L^−1^	[53]
	29.95772°	−9.79462°	2009	In situ	20–24	-	-		-	-	
Venezuela											
*Ostreopsis* cf. *siamensis*	10.59090°	−66.07386°	2010–2015	In situ	-	-	-	-	-	3596 cells L^−1^	[122]
Pacific and Indo-Pacific waters											
Vietnam											
*Ostreopsis siamensis*	12.18200°	109.29200°	2020	In situ	24–30	31–34	-	-	-	14,790 cells 100 cm^−2^	[45]
New Zealand											
*Ostreopsis siamensis*	−36.58878°	175.89378°	2004	In situ	21	-	-	-	-	1.4 × 10^6^ cells g^−1^ FWM	[14]
	−34.63077°	173.55539°	1995–1997	In situ	15.3–22.5	-	-	-	-	3.6 × 10^2^ cells g^−1^ FWM	[55]
	−35.29552°	174.53340°	1997–1999	GP Medium	18	-	100	Toxic for *Artemia salina*	0.3	-	[56]
Malaysia											
*Ostreopsis siamensis*	5.28278°	103.22669°	2015–2018	f/2 Si	25.0	30	90	Toxic	-	-	[119]
French Polynesia											
*Ostreopsis siamensis*	−17.64046°	−149.61025°	2019	f10k	26	36	60	Toxic	-	-	[46]
Japan											
*Ostreopsis siamensis*	24.43447°	124.28901°	-	-	-	-	-	-	-	-	[47]
*Ostreopsis* cf. *siamensis*	43.066736°	131.94946°	2010	In situ	18	30–34	-	-	-	52 × 10^3^ cells g^−1^ FWM	[123]
Australia											
*Ostreopsis* cf. *siamensis*	−23.44213°	151.91069°	-	-	-	-	-	-	-	-	[120]
	−24.11266°	152.71095°									
*Ostreopsis* sp. 9	−36.88330°	149.91671°	2013	f/2, f/10	18.0	35.0	60–100	Toxic	0.39	2.37 × 10^4^ cell mL^−1^	[114]

## Data Availability

The raw data supporting the conclusions of this article will be made available by the authors on request.

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
