# Peer review of "Molecular Phylogeny, Morphology, Growth and Toxicity of Three Benthic Dinoflagellates Ostreopsis sp. 9, Prorocentrum lima and Coolia monotis Developing in Strait of Gibraltar, Southwestern Mediterranean"

_toxins, 2024, doi:10.3390/toxins16010049_

Round 1
Reviewer 1 Report
Comments and Suggestions for Authors
See the attached pdf

Comments on the Quality of English LanguageThe English is in general very good, there are just a few spelling errors.
Author Response
29th November 2023
Responses to the Referee 1
Dear Referee, thank you very much for the corrections and suggestions you have provided which help us to improve our manuscript. All of your questions have been addressed, and the necessary clarifications and modifications have been implemented in the new version of manuscript. Our responses are in blue in the following text.
General Comments
Growth rate calculations – more detail is needed for how the growth rate calculaons were made as there is no supporng informaon provided for the stascal results presented. That is at the very least the number of data points used should be included. Currently there is a lot of contradictory informaon in the manuscript between the provided figures and tables as to what the actual growth rates were.
In the revised version of the manuscript, we have incorporated additional details regarding the calculation of the growth rate, following the methodology outlined by Guillard. Specifically, we have included information about the number of data points used to calculate the growth rate. To perform cell counting, we conducted measurements in triplicate cultures (three flasks with three counts for each flask/culture). A 5 mL sample was taken from each culture after homogenization in 250 mL Erlenmeyer flasks. These measurements were taken at the same time of day every three or four days. Furthermore, we have identified and rectified an inconsistency between the growth rate value presented in the figure and text, as compared to that mentioned in the table 2. The corrected growth rate value for a temperature of 24 °C is now accurately reported as 0.16±0.01 d-1.
In this context it would be more helpful for the reader if the growth rate data was ploted on a natural log scale so that the growth rate itself could be viewed. Currently in Figure 7 most of the key pieces of data have some clear inconsistencies in them (high count error, extremely long lag phase)
The application of natural logarithm to the algal concentrations renders it difficult to visually discern the lag, exponential and stationary growth phases. Their patterns were interesting to discuss and to compare between the three BHABs species. In Figure 7, the error bars appear comparatively low in contrast to similar studies found in the literature. It is worth noting that there are only two data points, specifically on the 19th and 21st days, which display some degree of variability in the growth of Ostreopsis within the three flasks/cultures monitored. It is crucial to underscore that this variability does not exert a substantial influence on the calculation of the growth rate and the mean calculated value, as the growth rate is computed using the entire data acquired during the exponential growth phase. The Standard deviation of the mean growth value remains low.
In this study there is also a strong indicaon that Prorocentrum lima was either strongly limited by the presence or absence of something which resulted in slow growth. One possibility could be Cu contaminaon of the seawater.
One of the explanation of the sluggish growth of P. lima, characterized by an extended lag phase, could be the limited initial cell concentration at the time of inoculation (530 cells.-1 at t0) but also the relatively lower growth rate of this species in comparison to Ostreopsis and Coolia, as depicted in Figure 8. This has now been introduced into the discussion.
It is noteworthy that we employed natural seawater collected from a station situated in the open waters of the Alboran Sea, far removed from potential contamination sources such as rivers or ports. Furthermore, we maintained consistency by utilizing the exact same seawater source for the cultivation of the other BHABs species under investigation namely Ostreopsis and Coolia as well as for planktonic dinoflagellates as Alexandrium which all showed noticeable growth rates.
Mixotrophy/phagotrophy in algal species – there is no menon in the manuscript as to whether the cell cultures were axenic or not. Though recent work on these species regarding the potenal for mixotrophy via phagotrophy on bacteria, is mixed [Almada et al., 2017; Lee and Park, 2018] it does indicate that these species may be able to feed heterotrophically.
Indeed, the cultures were not axenic, and this information has been included now in the Materials and Methods section (Section 5.2) as follows: "Three monoclonal non-axenic cultures were established...". We concur with the referee's suggestion that the BHABs species under investigation could potentially exhibit mixotrophic behavior through osmotrophy and/or phagotrophy mechanisms. The references which were provided by the referee are added in the version of the MS with the corresponding text. Please see below:
“Nitrogen is essential in the metabolism of phytoplankton as it’s a key element in proteins synthesis and the photosynthesis metabolism [138]. Many dinoflagellates exhibit mixotrophic behavior in their feeding strategy including osmotrophy and phagotrophy (Almada et al.2017; Stocker et al. 2017, Lee and Park 2018).”
Specific Comments ( sp – denotes spelling error)
Line 195: The error bars in figure 8 don’t seem to reflect the uncertainty seen in figure 7. This is most notably for Ostreopsis, where figure 7 has large errors bars at the me of the most rapid growth so the overall data would be more likely beter fited by a slower growth rate. In general the growth rate
Regarding the growth of Ostreopsis as shown in Figure 8, we acknowledge that there were only two data points with relatively high error bars, specifically on days 19 and 21. It is important to emphasize that the calculation of the growth rate was primarily based on data points acquired during the exponential growth phase, spanning from day 9 to day 19. We have calculated the growth rates using data obtained from three culture flasks. The mean value of the growth rate with its error bar shown in figure 8 is indicated now correctly into the text and the figure.
Line 196: This likely indicates that the culture was growth limited by some missing nutrient and it took some me for this to be synthesized or obtained from sources within the culture. In this regard it could be that the organisms were exisng by phagotrophy – see also the general comment above.
When examining previously published works, it appears that the growth of our Ostreopsis strain have comparable reported growth rates. Nonetheless, we concur with the referee's suggestion that this species might possess mixotrophic capabilities including phagotrophy, which could potentially enhance its growth. This aspect has been incorporated into the new version of the manuscript and is now discussed in the light of the added references you provided to us in regards to the phagotrophic nutrition mode of some dinoflagellate species.
Line 213: Please indicate the temperature and light condions used here, by including in the figure legend. Were other temperatures employed for Ostreopsis and Coolia?
Done
Line 235: The growth rate at 24°C for Prorocentrum lima appears to be around 0.08 here but is shown as being around 0.10 in figure 8, can this apparent difference be explained? Also was there similar lag phases of a 100 days seen in the experiments at different temperatures?
Thank you for addressing the issue. The corrected value of 0.16±0.1 day-1 has been appropriately updated in Table 2 and throughout the manuscript.
Furthermore, the extended lag phase observed at a temperature of 24°C can potentially be attributed to the initial cell concentration used for the inoculum, which was only 530 Cells/L for P. lima. It is well-established that the length of the lag phase tends to decrease as the cell concentration at t0 is increased. This information provides valuable insights into the observed growth patterns. Yes, we have long lag phases with other temperatures, data are not presented in this work. Yes, there was close lag phase for the other tested temperatures with the same concentration at the moment of the inoculation (t0) of P. lima cells.
Line 239: What happened to the urea experiments menoned in the methods secon?
We conducted preliminary experiments with urea; however, we encountered issues with the cultures, which is why the results are not presented in this manuscript. We intend to conduct new experiments in the coming months.
The sentence in the M&M section has been revised in the new version of the manuscript, as shown below:
“In order to test the effect of nitrogen on PLCM17 toxin content, the modified L1 and ENSW medium containing various sources of nitrogen; Nitrate (NaNO3) and Ammonium (NH4Cl) were prepared with three concentrations (441, 882 and 1764 µM) of each nitrogen form. Concentrations were chosen by multiplying by two and dividing by two the concentration of the original L1 medium.”
Line 245: (sp) growth
Done
Line 255: Figure 12 Part B and ‘C’ (sp) Staonary phase. Note also there are two part B’s in this figure and no part C.
Done
Line 255: Figure 12 Please include the error bars for part D of this figure
Done
Line 257: Please include the temperature, salinity and light condions employed here.
Done
Line 359: A slow growth rate leads to a ‘prolonged exponenal phase’, so this is hardly surprising, what is more strange is the extremely long lag phase.
Indeed, the length of the lag phase is influenced by the inoculum (the cell density at t0), and it can also be attributed to the relatively low growth rate of benthic dinoflagellates. We have taken special care to thoroughly address this aspect in the new version of the manuscript, particularly in the Discussion section
Line 430: This is not the usual use of the term ‘mixotroph’ [Stoecker et al., 2017; Ward, 2019; Wilken et al., 2019]. Strongly suggest that this sentence then be changed to reflect the diversity of nitrogen sources and the potenal as outlined above for heterotrophic growth.
The sentence has been modified to replace "mixotroph" with "osmotroph" to convey the capacity to utilize organic nitrogen compounds:
“Many dinoflagellate species exhibite mixotrophic beahvior in their feeding strategy including osmotrophy and phagotrophy (Almada t al. Stocker et al.). It has been demonstrated that some Prorocentrum species can use ammonium, nitrate, urea, and amino acids for its nitrogen supply which suggest that species of this genera use osmotrophy for thei nitrogen supply [129,139]”.
Line 445: (sp) surprisingly
Done
Line 546: What was the salinity of the natural seawater (NSW) and how and where was it obtained from? It’s possible that there were elevated levels of some trace metals (e.g. Cu, Sn) already in the NSW and this caused the long lag phase in the Prorocentrum lima incubaons [Gu et al., 2019; Rhodes et al., 2006]. Were any measurements made of the metal content of the NSW?
The salinity of the natural seawater used in our experiments was 36 and it was stable for our laboratory experiments. The seawater utilized to prepare our culture medium was collected from the Alboran Sea, located far from any terrestrial contamination sources (rivers, ports..).
Yes we agree with the referee that elevated of some trace elements could affect negatively the growth of dinoflagellates. However, in our study, we believe that the extended lag phase observed can primarily be attributed to two factors: the initial cell concentration, which was approximately 500 cells/ml in our experiments, and the inherently slow growth rate of BHABs species in particular P. lima, as demonstrated in numerous previous studies (refer to Bengharbia et al., 2016, and references therein). Furthermore, it is noteworthy that the same seawater was employed for our laboratory experiments involving other BHABs species and either planktonic dinoflagellates species of the Alexandrium genus and certain diatoms. These organisms exhibited high growth rates, suggesting the absence of trace metal contaminants in the seawater used for our experiments
Line 548: Axenic cultures?
The sentence was re-written to be more precise in the new version of MS.
"Three monoclonal non-axenic cultures were established"
Line 553: Why was the switch to L1 media made for Prorocentrum lima? Was it related to poor growth rates in the ENSW? Is all the data reported here for that species from L1 media? Did the L1 sll contain silicate? L1 has a very high N:P rao (24.4) – non Redfield- , so was the P content of the L1 media kept at the original concentraon for all of the nutrient experiments (lines 689-694)?
In our laboratory, we typically employ L1 medium for cultivating a wide range of dinoflagellate species, including P. lima. Initially, we attempted to culture Ostreopsis and Coolia in L1 medium for the purpose of comparing their growth with that of P. lima. Unfortunately, we encountered an issue where the cells of Ostreopsis and Coolia exhibited deformities, rendering it impossible to continue the cultures under these conditions. As a result, we resorted to using ENSW medium for Ostreopsis and Coolia in our experiments.
L1 medium does not contain silicate. We ensured that the concentration of phosphate was consistent across all experiments involving P. lima.
.Line 560: (sp) Lugol’s soluon.
Done
Line 674: Was it really 155 days for all temperature treatments conducted with P. lima?
Indeed, the highest cell densities of P. lima, or cell yield, were achieved after a period spanning from at least 100 to 155 days. This prolonged duration can be attributed to the combination of the dinoflagellate's inherently low growth rate and the limited size of the inoculum used in the culture at t0
Line 677-681: The definion of the exponenal phase is obviously crical here as some of the growth rates appear to be calculated from only 3 points which likely does not reflect the true error. The formula should also be reported as µm= (ln(N1)-Ln(N0))/(T1-T0) as currently as it does not include the required brackets the formula is incorrect. Lastly there is no linear poron of the exponenal growth phase, rather it is the exponenal poron of the growth phase.
We agree with the referee. We used the method of Guillard et al. which is currently used in the calculation of growth rate of the dinoflagellates. The corresponding paragraph is re-written completely to meet the requirement of the referee. Please see below :
“Cell concentrations of the three BHABs species under study were monitored in triplicate cultures. To perform cell counting, a 5 mL sample was taken after homogenization in 250 mL Erlenmeyer flasks (one separate culture in each flask) at the same time of day every three or four days. This monitoring continued for 28 days for O. cf. siamensis, 38 days for C. monotis, and 155 days for P. lima. The collected samples were fixed and then placed (1 mL each) on the Sedgewick-Rafter slide. Cell counting was conducted using an inverted photonic microscope, following the method outlined by Guillard [183]. To calculate the growth rate (expressed in day-1), a linear regression was performed over the entire exponential growth phase. This was achieved by applying the least square fit of a straight line to the data after logarithmic transformation. The formula used for calculating the growth rate was as follows: µm = (Ln(N1) - Ln(N0))/(T1 - T0), where µm represents the growth rate in units of day-1 ,N1 and N0 de-note the cell density at time T1 and T0, respectively, during the linear portion of the exponential growth phase.”
Line 690: What happened to the urea experiments? Also note that urea contains 2 equivalents of N and so should be used at half the concentraons in order to match for the N contribuon.
Yes, investigating the impact of urea on the growth of P. lima is very interesting as P. lima could exhibit osmotrophy for its nitrogen uptake and assimilation. We conducted preliminary experiments with urea; however, we encountered issues with the cultures, which is why the results are not presented in this manuscript. We intend to repeat these experiments in the coming months. They are really time consuming laboratory experiments.
We appreciate the mention of the interesting references you provided in regards to the different ways of dinoflagellate’s nutrition including mixotrophy. Some of the relevant references have been incorporated into the new version of the manuscript in both the discussion section of our results.
References
Almada, E. V. C., W. F. d. Carvalho, and S. M. Nascimento (2017), Investigation of phagotrophy in natural assemblages of the benthic dinoflagellates Ostreopsis, Prorocentrum and Coolia, Brazilian Journal of Oceanography 65(3), 392-399.
Gu, S., S.-W. Xiao, J.-W. Zheng, H.-Y. Li, J.-S. Liu, and W.-D. Yang (2019), ABC Transporters in
Prorocentrum lima and Their Expression Under Different Environmental Condions Including Okadaic Acid Producon, Marine Drugs, 17(5), 259.
Lee, B., and M. G. Park (2018), Genetic Analyses of the rbcL and psaA Genes From Single Cells Demonstrate a Rhodophyte Origin of the Prey in the Toxic Benthic Dinoflagellate Ostreopsis, Frontiers in Marine Science, 5.
Rhodes, L., A. Selwood, P. McNabb, L. Briggs, J. Adamson, R. van Ginkel, and O. Laczka (2006), Trace metal effects on the producon of biotoxins by microalgae, African Journal of Marine Science, 28(2), 393-397.
Stoecker, D. K., P. J. Hansen, D. A. Caron, and A. Mitra (2017), Mixotrophy in the Marine Plankton, Annual Review of Marine Science, 9(1), 311-335.
Ward, B. A. (2019), Mixotroph ecology: More than the sum of its parts, Proceedings of the National Academy of Sciences, 116(13), 5846-5848.
Wilken, S., C. C. M. Yung, M. Hamilton, K. Hoadley, J. Nzongo, C. Eckmann, M. Corrochano-Luque, C. Poirier, and A. Z. Worden (2019), The need to account for cell biology in characterizing predatory mixotrophs in aquatic environments, Philosophical Transactions of the Royal Society B: Biological Sciences, 374(1786), 20190090.

Reviewer 2 Report
Comments and Suggestions for Authors
The manuscript entitled “Molecular phylogeny, morphology, growth and toxicity of three benthic dinoflagellates Ostreopsis cf. siamensis, Prorocentrum lima and Coolia monotis developing in Gibraltar Strait, South Western Mediterranean” reports the isolation of three potentially harmful benthic dinoflagellate species from the Gibraltar Strait and their characterization in terms of shape, growth and toxin production under various parameters such as temperature, growth phase and nitrogen source. The work was meticulously planned, and experiments performed correctly; however, the manuscript still requires extensive revision before it is suitable for publication.
Major issues.
Lines 164-165: None of the reported sequences were successfully retrieved from the provided GenBank accession numbers. Will they be released after publication?
Figures 13, 14, 15 and Table 3 are more suitable if provided as supplementary files.
References are not numbered in order of appearance: for example, line 37 has [34,37,118]; line 40 has [25,39-41,160,161,163,165], [165, 168], then jumps to [181,185] on line 43, and so on. Consequently, no check for rigorous correspondence from list to text and vice versa was possible.
Minor issues and grammatical corrections.
Line 29: What does “Temperate waters know the proliferation of….” mean?
Line 38: Abbreviation of palytoxin is given as “PlTX” but “PLTX” is used subsequently; please correct.
Line 40: The statement starting with “Although…” makes no sense since the sentence never states what happens although.
Line 50: Use “PLTX-like”.
Line 61: The meaning of abbreviation “LST” has not been described; please indicate meaning.
Lines 61-62: Something missing in “…altering cellular physiological (what?)…”.
Line 88: LST abbreviation should be defined previously on line 61 and used abbreviated here.
Lines 92-93: The DTX abbreviation has been defined previously on line 59 and it should be used only abbreviated here.
Lines 97-99: This statement is confusing “The objective of this study was to characterize for the first time morphologically, genetically, and toxically three BHABs species (O. cf. siamensis, P. lima, C. monotis) isolated in the Gibraltar Strait and their monoclonal cultures established.”. Do the authors mean that their objective was to do their characterization on both, the isolated BHABs as well as on the established monoclonal cultures? Or the objectives were to isolate, establish the monoclonal cultures, and characterize them once in established cultures? Please state clearly.
Line 101: Following the above query the sentence must end with “…P. lima cultures.”.
Line 110: There is no plate with “7’’ visible in the figure; please mention the reason why.
Lines 160, 211 and 554: “Add space between “12” and “h” to read “12 h:12 h”.
Line 185: Abbreviation already provided for palytoxins at this point; only PTX should be used.
Line 188: Correct to “presence of Okadaic acid”; also the “OA” in parenthesis does not match the mark on the figure “(AO)”.
Line 189. Abbreviation for dinophysistoxins already provided; only “DTX-1” should be used.
Line 213-215: The figure legend mentions the species in the same order as in the previous A, B and C growth curves but in this other figure, they are not in the same order; please mention them in the corresponding order and indicate the color of the corresponding bar.
Lines 227-228: The table legend should mention the growth temperature used for the tests at various nitrate and ammonia concentrations.
Line 236 and 256: Figure 10 and 12 legends, respectively; please define “E” and “S” for (D).
Line 239: Here the growth temperature should also be mentioned at the beginning of the paragraph.
Line 248: Correct typo to “respectively”.
Line 286: Place period after quote sign.
Lines 305, 509: “Ostreopsis” should be in italics.
Lines 316, 319, 320: Use PTX for “palytoxin”.
Line 324: “Artemia” should be in italics.
Lines 398, 426: “P. lima” should be in italics.
Lines 401, 410, 429, 444: “Prorocentrum” should be in italics.
Line 415: Correct typo to “exhibited”.
Line 416: Correct typo to “did not exceed”.
Line 419: Correct typo to “was shown”.
Lines 435, 441: Mention the growth temperature for P. lima grown on nitrate or ammonia.
Line 443: Correct typo to “does not”.
Lines 445, 446: Correct typos to “surprisingly” and “stationary”.
Lines 444-456: The statement “Here, in presence of ammonia, the concentrations of OA and DTX-1 in the exponential phase are not high but surprisingly the LSTs levels increased reaching relatively high values in the stationary phase with the two ammonium concentrations 441 μM and 882 μM.” is hard to follow; rewrite in shorter sentences and more concisely.
Line 500: Delete “s” from “Gibraltars”.
Line 518: “Dinophysis” should be in italics.
Line 547: Close parenthesis.
Line 568: Use “g” instead of “RPM”; the “g” in italics.
Lines 568, 622: Use “min” instead of “minutes”.
Line 597: Change “was used” to “were used”.
Line 619: “g” must be in italics.
Line 645: Is a “1” missing in “DTX-“?
Line 685: Is the verb “was” missing to read “medium was inoculated”?
Line 690: Use the plural “sources”.
Line 696, 700: Here the g force is expressed as “x g” whereas previously only as “g”; use only one form always and italics for “g”.
Other issues.
Throughout Material and Methods use spaces between numbers and units, i.e. “1.5 mL”, “12 h”, etc., and use appropriate abbreviations “min”, “d”, etc.
Many references have genus and species names in normal font and must be changed to italics.
Comments on the Quality of English LanguageEnglish is acceptable and only minor corrections are required as mentioned above.
Author Response
29th November 2023
Responses ton referee 2
Comments and Suggestions for Authors
The manuscript entitled “Molecular phylogeny, morphology, growth and toxicity of three benthic dinoflagellates Ostreopsis cf. siamensis, Prorocentrum lima and Coolia monotis developing in Gibraltar Strait, South Western Mediterranean” reports the isolation of three potentially harmful benthic dinoflagellate species from the Gibraltar Strait and their characterization in terms of shape, growth and toxin production under various parameters such as temperature, growth phase and nitrogen source. The work was meticulously planned, and experiments performed correctly; however, the manuscript still requires extensive revision before it is suitable for publication.
Dear Referee, we sincerely appreciate your valuable comments, suggestions, and corrections, as they have significantly contributed to the enhancement of our manuscript. Below, you will find our responses, in blue and within the revised version, you will observe all the modifications made in accordance with your recommendations.
Major issues.
Lines 164-165: None of the reported sequences were successfully retrieved from the provided GenBank accession numbers. Will they be released after publication?
The sequences references have been integrated into the tree. They will be available on the first November 2024 but the date can be modified if necessary.
Figures 13, 14 and Table 3 are more suitable if provided as supplementary files.
From the outset of drafting this paper, we observed a substantial volume of data and articles pertaining to the geographical distribution, growth characteristics, and toxicity of Ostreopsis cf. ovata. In contrast, there was a scarcity of data, a degree of confusion in the reported strains, and a need for synthesizing information (regarding distribution, growth characteristics, toxicity, etc.) for the species Ostreopsis siamensis and Ostreopsis cf. siamensis. This aspect adds originality to our work and is likely to be of interest to researchers working on Ostreopsis genera. The data synthesis process spanned several months, involving a thorough examination of numerous publications. We wish to retain Figures and Table 3 within the main body of the manuscript. We really appreciate your understanding in this matter.
References are not numbered in order of appearance: for example, line 37 has [34,37,118]; line 40 has [25,39-41,160,161,163,165], [165, 168], then jumps to [181,185] on line 43, and so on. Consequently, no check for rigorous correspondence from list to text and vice versa was possible.
Thanks for your careful check of the references. We checked them rigorously in the text and in the list and proceeded to the necessary modifications.
Minor issues and grammatical corrections.
Line 29: What does “Temperate waters know the proliferation of….” mean?
The sentence is reformulated in the new version of the MS as “Numerous BHABs species, renowned for their thermophily, are increasingly proliferating within temperate marine ecosystems [11–17]”.
Line 38: Abbreviation of palytoxin is given as “PlTX” but “PLTX” is used subsequently; please correct.
Done
Line 40: The statement starting with “Although…” makes no sense since the sentence never states what happens although.
Although is deleted
Line 50: Use “PLTX-like”.
Done
Line 61: The meaning of abbreviation “LST” has not been described; please indicate meaning.
Done. In the new version of MS, LSTs was specified “The produced Lipophilic Shellfish Toxins (LSTs) are efficient…..”
Lines 61-62: Something missing in “…altering cellular physiological (what?)…”.
The sentence has been re-written with more accurate information on the effect of LSTs
“The produced Lipophilic Shellfish Toxins (LSTs), OA and DTXs are potent inhibitors of protein phosphatases 2A, 1B, and 2B, which may promote cancer in the human digestive system”
Line 88: LST abbreviation should be defined previously on line 61 and used abbreviated here.
Done
Lines 92-93: The DTX abbreviation has been defined previously on line 59 and it should be used only abbreviated here.
Done
Lines 97-99: This statement is confusing “The objective of this study was to characterize for the first time morphologically, genetically, and toxically three BHABs species (O. cf. siamensis, P. lima, C. monotis) isolated in the Gibraltar Strait and their monoclonal cultures established.”. Do the authors mean that their objective was to do their characterization on both, the isolated BHABs as well as on the established monoclonal cultures? Or the objectives were to isolate, establish the monoclonal cultures, and characterize them once in established cultures? Please state clearly.
The sentence was re-written to remove any confusion in the text.
“The aim of this study was to comprehensively characterize, for the first time, three BHABs dinoflagellates (O. cf. siamensis, P. lima, C. monotis) isolated in the Gibraltar Strait, focusing on their morphology, phylogeny, and toxicity. Furthermore, we examined the impact of increasing temperature and varying concentrations of ammonia and nitrate on the growth, biometry, and toxin levels within P. lima cultures.”
Line 101: Following the above query the sentence must end with “…P. lima cultures.”.
Done
Line 110: There is no plate with “7’’ visible in the figure; please mention the reason why.
We’ve changed the contrast and color to make the figure 1C easier to visualize
Lines 160, 211 and 554: “Add space between “12” and “h” to read “12 h:12 h”.
Done
Line 185: Abbreviation already provided for palytoxins at this point; only PTX should be used.
Done
Line 188: Correct to “presence of Okadaic acid”; also the “OA” in parenthesis does not match the mark on the figure “(AO)”.
Done. Of is added.
The mark in the corresponding figure 6 is also corrected and it is now OA (Okadaic Acid).
Figure 6. Chromatogram of LC–MS/MS analysis of PLCM17 Prorocentrum lima strain from Gibraltar strait, Southern Mediterranean. OA and DTX-1 mean Okadaic Acid and DinophysisToxin-1, respectively.
Line 189. Abbreviation for dinophysistoxins already provided; only “DTX-1” should be used.
Done
Line 213-215: The figure legend mentions the species in the same order as in the previous A, B and C growth curves but in this other figure, they are not in the same order; please mention them in the corresponding order and indicate the color of the corresponding bar.
Done.
Lines 227-228: The table legend should mention the growth temperature used for the tests at various nitrate and ammonia concentrations.
The legend has been modified:
“Figure 11. Okadaic acid and DTX-1 concentrations during exponential and stationary growth phases of Prorocentrum lima PLCM17 strain cultivated at different concentrations of nitrate and ammonia. Cultures were grown at a temperature of 24 °C, salinity of 36 and irradiance of 90 µmol photons.m-2.s-1 with 12 h : 12 h dark/light cycle.”
Line 236 and 256: Figure 10 and 12 legends, respectively; please define “E” and “S” for (D).
The legend has been corrected
“Figure 12. Growth rate in d-1 (A), cell width (B) and length (C) in µm and length/width ratio (D) measured at exponential and stationary growth phases of Prorocentrum lima PLCM17 strain culti-vated at different concentrations of nitrate and ammonia. Cultures were carried out at a temperature of 24 °C, salinity of 36 and irradiance of 90 µmol photons.m-2.s-1 with 12 h : 12 h dark/light cycle. In Fig 12D, E and S refer to Exponential and Stationary growth phases, respectively.”
Line 239: Here the growth temperature should also be mentioned at the beginning of the paragraph.
Done. Please, see the new sentence in the revised version of the MS.
“For nitrogen experiments, cultures were grown at temperature of 24 °C. Data showed that LSTs concentrations were slightly higher in the exponential phase….”
Line 248: Correct typo to “respectively”.
Done
Line 286: Place period after quote sign.
Done
Lines 305, 509: “Ostreopsis” should be in italics.
Done
Lines 316, 319, 320: Use PTX for “palytoxin”.
Done
Line 324: “Artemia” should be in italics.
Done
Lines 398, 426: “P. lima” should be in italics.
Done
Lines 401, 410, 429, 444: “Prorocentrum” should be in italics.
Done
Line 415: Correct typo to “exhibited”.
Done
Line 416: Correct typo to “did not exceed”.
Done
Line 419: Correct typo to “was shown”.
Done
Lines 435, 441: Mention the growth temperature for P. lima grown on nitrate or ammonia.
For nitrogen experiments, P. lima was cultivated at 24 °C. The sentence was changed to bring this information that was already given in the M&M. Please see below.
“In our experiments with nitrate and ammonia as nitrogen sources, P. lima was grown at a temperature of 24 °C. In the presence of nitrate, OA and DTX-1 levels decreased when the concentration of this nitrogen was 441 µM and particularly in the stationary phase.”
Line 443: Correct typo to “does not”.
Done
Lines 445, 446: Correct typos to “surprisingly” and “stationary”.
Done
Lines 444-456: The statement “Here, in presence of ammonia, the concentrations of OA and DTX-1 in the exponential phase are not high but surprisingly the LSTs levels increased reaching relatively high values in the stationary phase with the two ammonium concentrations 441 μM and 882 μM.” is hard to follow; rewrite in shorter sentences and more concisely.
To bring more clarity to the sentence, it has been rewritten.
“ Our results showed that in the presence of ammonia, the concentrations of OA and DTX-1 in P. lima cells are higher in the stationary phase of growth in comparison to those measured in the exponential phase.”
Line 500: Delete “s” from “Gibraltars”.
Done
Line 518: “Dinophysis” should be in italics.
Done
Line 547: Close parenthesis.
Done
Line 568: Use “g” instead of “RPM”; the “g” in italics.
Values in RPM were converted in values in g, in accordance with the technical specifications of the centrifuge. g now is in italic.
14000 RPM is replaced by 17900 g
Lines 568, 622: Use “min” instead of “minutes”.
Done
Line 597: Change “was used” to “were used”.
Done
Line 619: “g” must be in italics.
Done
Line 645: Is a “1” missing in “DTX-“?
Done
Line 685: Is the verb “was” missing to read “medium was inoculated”?
Indeed, you are correct. Thank you.
“Sterile flasks, each containing 250 ml of culture medium, were inoculated with cells harvested during the exponential growth phase of each strain, which had been cultivated under identical conditions previously.”
Line 690: Use the plural “sources”.
Done
Line 696, 700: Here the g force is expressed as “x g” whereas previously only as “g”; use only one form always and italics for “g”.
g force is expressed now only as g
Other issues.
Throughout Material and Methods use spaces between numbers and units, i.e. “1.5 mL”, “12 h”, etc., and use appropriate abbreviations “min”, “d”, etc.
Done
Many references have genus and species names in normal font and must be changed to italics.
Done

Round 2
Reviewer 1 Report
Comments and Suggestions for Authors
The authors have made a good effort in replying to the earlier comments on their manuscript.
There is still the outstanding issue of reproducibility and the slow growth rates. As it is not clear from the manuscript if the slow growth rates continued upon re-innoculation of the cultures into fresh media. In particular it seems that the cultures were only grown once and thus the data is not reporting cells acclimated to the growing conditions. The slow growth rate of P. Lima in the initial stages also does not adequately describe the long lag phase at the start as the growth rate is still reported as 0.18 which implies a doubling every 3-3.5 days so in 35 says an increase in cell numbers of 2^10 (1024) would be expected, but there is no appreciable change in cell number. So likely they are not growing at all or the majority of the cells are non viable at the start. This non acclimation of the cells and lack of reproducibility needs to be discussed in the final version of the manuscript.
Comments on the Quality of English LanguageManuscript needs to be checked for English spelling - as there were some errors in the reply to the reviewer comments in the new text added to the paper.
Author Response
December 12th 2023
Responses to the referee
Comments and Suggestions for Authors
The authors have made a good effort in replying to the earlier comments on their manuscript.
We are pleased to have met the referee's requirements. Indeed, the corrections and suggestions made by the referee have greatly improved our manuscript, Thanks again.
There is still the outstanding issue of reproducibility and the slow growth rates. As it is not clear from the manuscript if the slow growth rates continued upon re-innoculation of the cultures into fresh media. In particular it seems that the cultures were only grown once and thus the data is not reporting cells acclimated to the growing conditions. The slow growth rate of P. Lima in the initial stages also does not adequately describe the long lag phase at the start as the growth rate is still reported as 0.18 which implies a doubling every 3-3.5 days so in 35 says an increase in cell numbers of 2^10 (1024) would be expected, but there is no appreciable change in cell number. So likely they are not growing at all or the majority of the cells are non viable at the start. This non acclimation of the cells and lack of reproducibility needs to be discussed in the final version of the manuscript.
As we discussed and as demonstrated by several authors and synthesized by Ben Gharbia et al. in 2016 in Toxins (in Figure 6, Table A2), the growth rate of Prorocentrum lima is generally low, and our results on PLCM17 strain form Gibraltar strait corroborated these findings.
The calculated growth rate, according to Guillard's method (1973), represents the maximum specific growth rate (µmax) calculated based on the exponential phase of growth. This precision has now been added to the Materials & Methods, Results and Discussion sections.
We consistently obtained reproducible P. lima growth rates in our laboratory experiments when cultivating this dinoflagellate under standard optimal conditions (24 °C, salinity of 36, and irradiance of 90 µmol.m-2.s-1). The calculated mean maximum growth rates for the PLCM17 strain ranged from 0.17 to 0.22 at a temperature of 24 °C.
In Ecophysiological controlled laboratory experiments, it is challenging and time-consuming to acclimate the algae to the tested conditions, such as when assessing the effects of different concentrations of nitrogen, phosphorus, increasing temperature, and salinity on the growth of the investigated algae. Nonetheless, in our study, we gradually acclimated P. lima PLCM17 strain to different tested temperatures (15, 20, 24, and 29 °C) by increasing or decreasing the temperature (24 °C) by 0.5 °C daily. In future studies, it will be interesting to acclimate P. lima to different tested concentrations of nitrogen, salinity, and phosphorus. This is now mentioned in M&M and discussed in the new version of the manuscript.
The modified and added text to respond to the referee requirement are highlighted in green in the new version of the MS.
Manuscript needs to be checked for English spelling - as there were some errors in the reply to the reviewer comments in the new text added to the paper.
The English spelling of the manuscript have been carefully checked.
However, if this is not enough, we can consider English editing by a recognized service.
